# Best of Three Worlds:
# Adaptive Experimentation for Digital Marketing in Practice

## ABSTRACT

Adaptive experimental design (AED) methods are increasingly being used in industry as a tool to boost testing throughput or reduce experimentation cost relative to traditional A/B/N testing methods. However, the behavior and guarantees of such methods are not well-understood beyond idealized stationary settings. This paper shares lessons learned regarding the challenges of naively using AED systems in industrial settings where non-stationarity is prevalent, while also providing perspectives on the proper objectives and system specifications in such settings. We developed an AED framework for counterfactual inference based on these experiences, and tested it in a commercial environment.

## KEYWORDS

Experimentation, multi-armed bandit, online decision-making

**ACM Reference Format:**
Anonymous Author(s). 2024. Best of Three Worlds: Adaptive Experimentation for Digital Marketing in Practice. In *The Web Conference '24: May 13–17, 2024, Singapore.* ACM, New York, NY, USA, 18 pages. https://doi.org/XXXXXXX.XXXXXXX

## 1 INTRODUCTION

A/B/N testing is a classic and ubiquitous form of experimentation that has a proven track record of driving key performance indicators within industry [14]. Yet, experimenters are steadily shifting toward *Adaptive Experimental Design* (AED) methods with the goal of increasing testing throughput or reducing the cost of experimentation. AED promises to use a fraction of the impressions that traditional A/B/N tests require to yield precise and correct inference or to directly drive business impact. However, the behavior and guarantees of these approaches are not well-understood theoretically nor empirically in commercial environments where idealized stationary feedback assumptions fail to hold. This paper aims to show real-world experiment data, bring attention to the unique challenges of using AED systems in production, and present a system designed to satisfy curated objectives with theoretical guarantees and proven empirical performance in production.

*Contributions.* Through simple, yet illustrative experimentation case studies, we bring to light key challenges to using AED systems effectively in practical, industrial settings. We demonstrate that existing estimation procedures and algorithmic methods can often

fail in these settings. The case studies are generalized to provide perspectives on the proper objectives and system specifications in these settings. We present the approach we have taken based on the lessons learned from working on AED systems in industrial settings over time, which identifies the counterfactual optimal treatment efficiently, mitigates opportunity cost, and is robust to the form of time variation we regularly observe (the *best of three worlds*). Our approach combines a cumulative gain estimator with always-valid inference and an elimination-based algorithmic approach. The experiments we present from our production system highlight how regret-minimizing algorithms such as TS, which assume stochastic environments, can fail spectacularly both for accruing a reward metric and for making inferences. In addition to empirical evidence, we also provide theoretical guarantees.

## 2 REAL-WORLD STUDIES AND LESSONS

We now present experimentation case studies and then generalize to discuss lessons learned using AED systems in industrial settings, along with their proper objectives and system specifications.

### 2.1 Case Study: Adaptive Designs & Inference

Imagine a setting where on a retailer web page, a marketer has been running a message $A$ for the last year and now wants to test whether message $B$ beats $A$. Fearful of incurring a large amount of loss from A/B testing opportunity cost, the marketer chooses to use an adaptive experimental method, namely TS. At the start of the experiment, the messages are initialized with a default prior distribution, and then at each round the bandit dynamically allocates traffic to each treatment, playing each message according to the posterior probability of its running empirical mean being the highest [19]. After day 8, the algorithm directs most traffic to message $A$ (see Figure 1). On day 14, the experimenter needs to decide whether $A$ has actually beaten $B$. They conduct a paired $t$-test which, somewhat surprisingly, does not produce a significant $p$-value. As the bandit shifted all traffic to message $A$, not enough traffic was directed to message $B$, diminishing the power of the test. The experimenter is forced to conclude that they can not reject the null hypothesis that there is no difference between the messages. A few days later, the experimenter, who is still perplexed, looks at the daily empirical means and is then shocked to see that on most days, $B$ tends to have a higher daily empirical mean than $A$, which disagrees with the bandit's beliefs that led to the traffic allocation it produced.

To understand this behavior, note that in Figure 1c the running empirical mean of $A$ is exceeding that of $B$, leading the algorithm to put all its traffic on $A$. This phenomenon where the running empirical mean shows a different direction than daily comparisons is known as *Simpson's Paradox*, and occurs in settings where the traffic is dynamically allocated to arms whose means change over time [14]. Intuitively, the experimenter has made a Type I error by

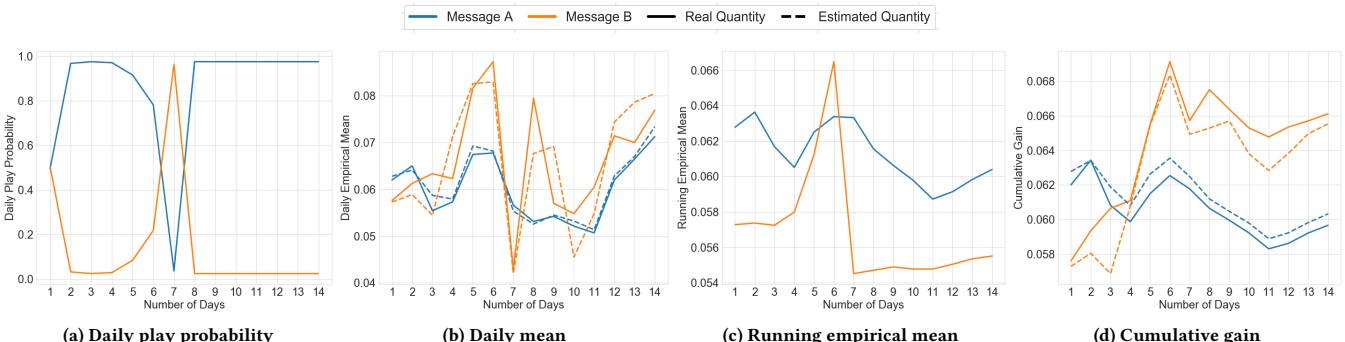

**Figure 1: Case study of time-variation and adaptive allocations causing Simpson's paradox.**

(a) Daily play probability    (b) Daily mean    (c) Running empirical mean    (d) Cumulative gain

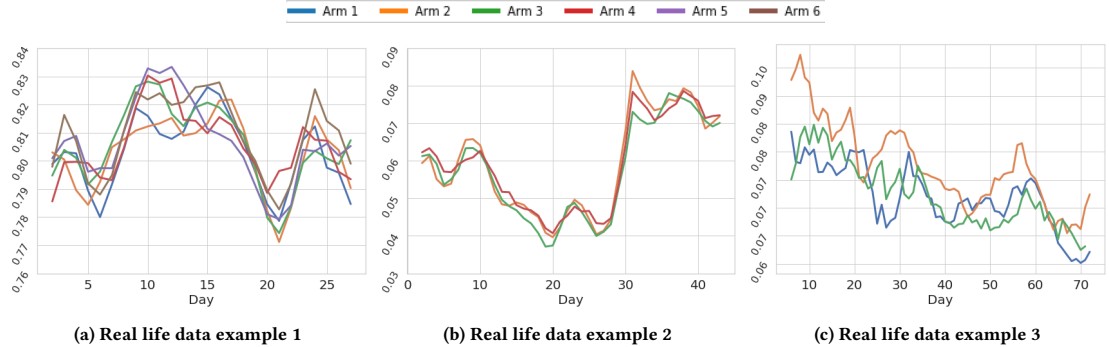

(a) Real life data example 1    (b) Real life data example 2    (c) Real life data example 3

**Figure 2: Daily empirical means from marketing experiments with uniformly collected data.**

trusting the algorithm and choosing arm $A$. Indeed, during the time period from days 8 to 14, the algorithm decided to put more traffic on arm $A$, exacerbating Simpson's paradox. Convinced by its own bad decision, the algorithm then chooses a bad traffic allocation which further exacerbates the problem and leads to a vicious cycle.

## 2.2 Case Study: Real Life Time Variation

The approach we develop in this paper is motivated by the type of data that is regularly observed in industrial settings. Typically, industrial data is not stationary, but it is also not fully adversarial. We show that the underlying data processes lie somewhere in between these extremes. Figure 2 shows the daily empirical means for each treatment within several marketing experiments where the data was collected uniformly at random between treatments. Clearly, significant non-stationarity in the underlying performance of all treatments in an experiment is the norm and not the exception. The data shows both trends and cyclicity, yet the structure is inconsistent within any given experiment as well as across experiments. Consequently, it is challenging to adopt solutions that model latent confounders a priori [16]. On the other hand, the variation in the performance gaps between treatments is more well-behaved as a function of time. Together, this case study reveals that the objective of identifying the counterfactual optimal treatment in an experiment is often well-defined.

## 2.3 Lessons Learned

We now dive deeper into the general challenges of using AED methods in practice raised by the case studies and provide thoughts on industrial objectives and specifications that guide our approach.

*Regret Minimization Isn't Enough.* The fundamental goal of experimentation is to test hypothesis and deliver results that allow for *future iterations* [14]. As a result, it is important that experimentation procedures give the experimenter the ability to arrive at valid and measurable inferences. In settings where the experimenter wants to learn the best treatment, optimal regret minimization procedures take a significantly longer time to return the identity of the best arm with high probability [2, 3] and lead to biased mean estimates [20].

*Be Wary of the Batch.* Most experimental systems use batched model updates (daily or weekly). In the example from Section 2, the traffic was not constant daily (not shown), so an update on one day can have an undue impact on the rest of the experiment time. In experiments over short horizons, this implies that observations on the first few days can have a disproportionate impact on the traffic allocation and also the subsequent inferences that are made.

*Stochastic Bandit Algorithms Often Fail.* While it is common in industrial systems to deploy regret-minimizing algorithms based on underlying stationarity assumptions, these algorithms fail with regularity in any given experiment even for the sole purpose of

accruing an optimization metric. Often these failures go unnoticed due to the absence of a suitable comparison to bring attention to the problem (an appropriate A/A test). We highlight in our experiments that stochastic bandit algorithms can fail to maximize the accumulation of an optimization metric as a result of dynamic traffic allocations in combination with an estimation based on the observed adaptively collected data in time-varying environments.

*Identify the Counterfactual Best.* In settings where arm means are shifting over time, it is challenging to define the notion of a "best-arm" as the mean performance of an arm and the identity of the best arm may change daily. To bridge this gap, our proposed objective is to *identify with high probability the treatment that would have obtained the highest possible reward, if all traffic had been diverted to it*. This counterfactual metric is known as the *cumulative gain*. Figure 1d demonstrates the cumulative gain over time for the case study. With the exception of [1], we believe that this objective has hardly been considered in the best-arm identification literature.

*Always Valid Inference.* In traditional A/B/N testing, the experiment horizon is fixed ahead of time with a significance test at the end of the experiment. Monitoring *p*-values computed during the experiment is heavily frowned upon as it leads to Type 1 error inflation [12]. Recent work in the experimental space has lead to generalizations of the *p*-value known as *always-valid p-values* that can safely be sequentially monitored [8, 9, 12, 17]. This capability is critical in practice to allow for early stopping with valid inferences.

**The Best of Three Worlds (BOTW).** Though optimal regret minimization procedures fail to provide valid inferences and tend to identify the best arm more slowly, we still would like to minimize the cost of experimentation. Thus experimentation systems should try to provide the best of three worlds: identification of the counterfactual best, mitigation of opportunity cost, and robustness to arbitrary time variation. In completely adversarial settings we can't hope to have all three [1], but real life settings mostly live somewhere between fully stochastic and fully adversarial.

## 3  OUR APPROACH

We now describe our approach for experimentation that provides the best of three worlds. The method combines always-valid inference on estimators which are robust to time variation with an elimination-based algorithmic approach.

### 3.1  Experimentation Setting

Let us focus on the example of displaying ads on an online service. To set notation for the *time-varying* or *non-stochastic* setting, we assume an experiment consisting of $k$ arms running for a period of $T$ days beginning on day $t = 1$. On any day $t \in [T]$, arm $i \in [k]$ receives $n_{i,t}$ impressions and $n_t = \sum_{i \in [k]} n_{i,t}$ is the total amount of traffic on that day.[1] The sample count $n_{i,t}$ for each arm $i \in [k]$ on any day $t \in [T]$ is a stochastic quantity based on the rewards and samples given to the arms up to day $t \in [T]$. We assume that the underlying behavior of an arm $i \in [k]$ on day $t \in [T]$ is fixed over the period of a day and described by a Bernoulli distribution with mean $\mu_{i,t} \in (0, 1)$. Finally, we let $r_{i,t}$ and $\widehat{\mu}_{i,t} := r_{i,t}/n_{i,t}$ denote the total reward and daily empirical mean on day $t \in [T]$ for any

---

[1]We adopt the standard set notation of $[n] = \{1, 2, \ldots, n\}$ for any $n \in \mathbb{Z}_+$.

arm $i \in [k]$, respectively. Thus, conditional on the allocation $n_{i,t}$, $r_{i,t} \sim \text{Binomial}(n_{i,t}, \mu_{i,t})$ for each arm $i \in [k]$ on any day $t \in [T]$.

### 3.2  Estimation with Time Variation

We now discuss estimation in the presence of time-variation.

*3.2.1  Running Empirical Means.* We begin by discussing the standard approach in the experimentation literature of using the running empirical mean estimator to assess performance given a set of data that has been collected. The running empirical mean of arm $i \in [k]$ after $T$ days of an experiment with the total traffic to the arm denoted by $\bar{n}_{i,T} := \sum_{t=1}^{T} n_{i,t}$ is:

$$\bar{\mu}_i := (\textstyle\sum_{t=1}^{T} r_{i,t})/\bar{n}_{i,T} = (\textstyle\sum_{t=1}^{T} n_{i,t}\widehat{\mu}_{i,t})/\bar{n}_{i,T}. \tag{1}$$

Given the standard assumptions of fixed horizon A/B/N testing, in which the performance of each arm $i \in [k]$ is fixed ($\mu_{i,t} = \mu_i \ \forall \ t \in [T]$) and where each arm receives a constant, pre-determined proportion of the traffic each day, the running empirical mean is an unbiased estimator of the underlying performance ($\mathbb{E}[\bar{\mu}_{i,T}] = \mu_i$).

However, when the underlying performance of an arm exhibits daily time-variation, the running empirical mean can be a problematic estimator. To begin, the estimate $\bar{\mu}_{i,T}$ is subject to *Simpson's Paradox* [14]. In the context of experimentation, Simpson's paradox refers to a circumstance in which the daily empirical mean of an arm $i \in [k]$ is higher than that of an arm $j \in [k]$ on each given day ($\widehat{\mu}_{i,t} > \widehat{\mu}_{j,t} \ \forall \ t \in [T]$), *but* the running empirical mean of arm $j$ is higher than that of arm $i$ over the course of an experiment ($\bar{\mu}_{j,T} > \bar{\mu}_{i,T}$). As we saw in Case Study 1 (Figure 1c), in experimentation where the traffic allocation is changing over time, this paradox regularly arises. Moreover, for argument, if we suppose that the allocation of impressions is predetermined but not necessarily constant, then for any $i \in [k]$, $\mathbb{E}[\bar{\mu}_{i,T}] = \sum_{t=1}^{T} (n_{i,t}/\bar{n}_{i,T})\mu_{i,t}$. Thus, in expectation, the running empirical mean estimator of Equation (1) is estimating a rather arbitrarily weighted sum of the daily means that *depends on the allocation*. Finally, even if the underlying means are stationary, the running empirical mean is a biased estimator when the traffic is being collected adaptively [4, 6, 20, 21]. These estimator problems are empirically demonstrated in Section 4.

*3.2.2  Cumulative Gain.* As the above discussion implies, the running empirical mean estimator has many negative characteristics that make it inappropriate for time-varying settings with adaptive traffic allocation. Part of the challenge is that in time-varying settings the notion of "the best performing arm" may be poorly defined since the best-arm may change from day-to-day. To overcome this, we instead try to answer the following counterfactual: **"how much reward would this arm have accrued if it had received all of the traffic"**. More precisely, for any arm $i \in [k]$, the cumulative gain (CG) after $T$ days is defined as

$$G_{i,T} := \textstyle\sum_{t=1}^{T} n_t \mu_{i,t}. \tag{2}$$

Let total experiment traffic count after $T$ days be $\bar{n}_T := \sum_{t=1}^{T} n_t$. The corresponding cumulative gain rate variant is:

$$\bar{G}_{i,T} := G_{i,T}/\bar{n}_T. \tag{3}$$

In stationary settings, the cumulative gain rate reduces to the underlying mean, that is, $\bar{G}_{i,T} = \mu_i$ for arm $i \in [k]$ when $\mu_{i,t} = \mu_i$ for all days $t \in [T]$. In general, the cumulative gain rate reduces to

a weighted average of the daily means. In particular, for any arm $i \in [k]$, the cumulative gain rate after $T$ days can be written as $\bar{G}_{i,T} = \sum_{t=1}^{T} w_t \mu_{i,t}$ where the weight $w_t := n_t / \bar{n}_T$ is the proportion of the total experiment traffic that came on day $t$.

*Cumulative Gain Estimator.* In a general experimentation setting, we can build an estimator for the cumulative gain metric with desirable properties using inverse probability weighting [7]. Assume that on each day $t \in [T]$ of the experiment, a probability vector $p_t = (p_{1,t}, \cdots, p_{k,t}) \in \Delta_k$ is chosen according to the history up to day $t$.[2] Then, each visitor $s_t \in [n_t]$ on day $t \in [T]$ is shown an arm $I_{s_t} \in [k]$ that is selected with probability $\mathbb{P}(I_{s_t} = i) = p_{i,t}$ and a corresponding reward $r_{s_t}$ is observed. A natural cumulative gain estimator is given by inverse propensity weighing:

$$\widehat{G}_{i,T} = \sum_{t=1}^{T} (r_{i,t}/p_{i,t}). \tag{4}$$

Proposition 1 establishes the estimator is unbiased (proof in App. A).

PROPOSITION 1. *For any arm $i \in [k]$ and day horizon $T$, the estimator $\widehat{G}_{i,T} = \sum_{t=1}^{T} (r_{i,t}/p_{i,t})$ is unbiased for the cumulative gain. That is, we have $\mathbb{E}[\widehat{G}_{i,T}] = G_{i,T}$ as defined in Equation (2).*

The cumulative gain estimator will never suffer from Simpson's paradox, unlike the running empirical mean estimator which is prone to this phenomenon. Indeed, if $\widehat{\mu}_{i,t} > \widehat{\mu}_{j,t}$ for all $t \in [T]$ for some pair of arms $i, j \in [k]$, then we necessarily have $\widehat{G}_{i,T} \geq \widehat{G}_{j,T}$. As shown in the case study, Figures 1c and 1d, using the cumulative gain would have prevented misleading inferences from Simpson's paradox. The cumulative gain metric and its corresponding estimator can seamlessly be used for the purpose of assessing and estimating the performance gap between arms by taking the difference between the quantities for a pair of arms. In particular, we have that $\mathbb{E}[\widehat{G}_{i,T} - \widehat{G}_{j,T}] = G_{i,T} - G_{j,T}$ for any pair of arms $i, j \in [k]$.[3] The cumulative gain is a familiar quantity from the non-stochastic bandit research field [1]. In particular, it is precisely the quantity that is being measured when computing regret in non-stochastic bandit problems and the basis of robust regret minimization algorithms.

3.2.3 *Always-Valid Inference.* Now that we have defined a performance metric and analyzed the properties of a corresponding estimator, we shift our focus to describing how the tools developed so far can enable robust inferences in experimentation. While fixed horizon statistics and corresponding hypothesis tests are commonly used in production systems for experimentation, they are subject to a high risk of abuse and error rate inflation through repeated evaluation of the outcomes by practitioners and business stakeholders [12]. This motivates adopting *always-valid confidence intervals* [8, 10, 12], which allow experiments to be sequentially monitored without inflation of the error rate.

We directly focus on the cumulative gain gap between a pair of arms since we are interested in using always-valid intervals for making inferences on the comparison of arms. In this context, an always-valid confidence interval $C(i, j, t, \delta)$ for a pair of arms $i, j \in [k]$ with error tolerance $\delta \in (0, 1)$ guarantees

$$\mathbb{P}(\exists\, t \geq 1, i, j \in [k] : |\widehat{G}_{i,j,t} - G_{i,j,t}| \geq C(i, j, t, \delta)) \leq \delta.$$

---

[2]The notation $\Delta_k$ is used to denote the $k-1$ dimensional simplex.

[3]We adopt the notation $G_{i,j,T} := G_{i,T} - G_{j,T}$ and $\widehat{G}_{i,j,T} := \widehat{G}_{i,T} - \widehat{G}_{j,T}$.

---

**Algorithm 1** Cumulative Gain Successive Elimination (CGSE)

1: **Input** Arm set $[k]$, error tolerance $\delta \in (0, 1)$
2: **Initialize** Active arm set $\mathcal{A} \leftarrow [k]$, day $t \leftarrow 1$
3: **while** $|\mathcal{A}| > 1$ **do**
4:     Set $p_{i,t} = 1/|\mathcal{A}|$ for all $i \in \mathcal{A}$ and $p_{i,t} = 0$ for all $i \in [k] \setminus \mathcal{A}$
5:     For each arrival $s_t \in [n_t]$ show arm $I_{s_t} \sim p_t$
6:     Collect observations $\{r_{s_t}, I_{s_t}, p_t\}_{s=1}^{n_t}$
7:     **Eliminate suboptimal arms:**
8:     $\mathcal{A} \leftarrow \mathcal{A} \setminus \{j \in \mathcal{A} \text{ s.t. } \exists\, i \in \mathcal{A} : \widehat{G}_{i,j,t} - C(i, j, t, \delta/k) > 0\}$
9:     $t \leftarrow t + 1$
10: Return $\mathcal{A}$

---

There are several possible ways to derive an always-valid confidence interval for this quantity. To obtain the always-valid confidence interval, we now apply the MSPRT[4] using the plugin estimators $\widehat{\mu}_{i,t}$ and $\widehat{\mu}_{j,t}$ for the unknown daily arm means $\mu_{i,t}$ and $\mu_{j,t}$ on each day in an estimate of the variance. As a result, under the stated assumptions, we have

$$\mathbb{P}(\exists\, t \geq 1, i, j \in [k] : |\widehat{G}_{i,j,t} - G_{i,j,t}| \geq C(i, j, t, \delta)) \leq \delta,$$

$$\text{with} \quad C(i, j, t, \delta) := \sqrt{(\widehat{V}_{i,j,t} + \rho) \log\left((\widehat{V}_{i,j,t} + \rho)/(\rho\delta^2)\right)} \tag{5}$$

where $\rho > 0$ is a fixed constant and

$$\widehat{V}_{i,j,t} = \sum_{\tau=1}^{t} n_\tau \left(\widehat{\mu}_{i,\tau}(1 - \widehat{\mu}_{i,\tau})/p_{i,\tau} + \widehat{\mu}_{j,\tau}(1 - \widehat{\mu}_{j,\tau})/p_{j,\tau}\right).$$

This characterization immediately allows for sequential monitoring of cumulative gain gap estimates through upper and lower bounds for high probability decision-making. For more details on the confidence interval justification see Appendix C.2.

Finally, we make two remarks about the variance of the cumulative gain estimator. First, in general since our cumulative gain estimator is based on inverse propensity weighting, if any of the arm allocation probabilities are very small, the estimator variance can become very large. In Appendix C.1 we discuss the bias/variance trade-off between the running empirical mean estimator and the cumulative gain estimator. Second, [11] propose a similar estimator to the cumulative gain estimator. However, they make the strong assumption that $\mu_{i,t} = \mu_i + \gamma_t$ for all $i \in [k]$ where $\gamma_t$ is an exogenous shock. For this restricted setting, we demonstrate in Appendix B the variance reduced cumulative gain (VRCG) estimator

$$\widehat{G}^*_{i,j,T} = \sum_{t=1}^{T} (\widehat{\mu}_{i,t} - \widehat{\mu}_{j,t}) w_t, \quad \text{with } w_t \text{ defined as}$$

$$w_t = \left(\sum_{t=1}^{T} (n_{i,t}^{-1} + n_{j,t}^{-1})^{-1}\right)^{-1} (n_{i,t}^{-1} + n_{j,t}^{-1})^{-1} \quad \forall\, t \in [T],$$

is the minimal variance weighted estimator for the difference of means $\mu_i - \mu_j$. VRCG thus outperforms the cumulative gain estimator or the estimator proposed in [11] in this restricted setting.

## 3.3 Adaptive Counterfactual Inference

We seek an AED method achieving the best of three worlds. That is an algorithm giving confident, sample efficient identification of the counterfactual optimal treatment, while simultaneously minimizing regret in experiments where stationarity is not guaranteed.

---

[4]For reference, see Eq. 14 in [8].

*Algorithm Description.* For adaptive counterfactual inference, we adopt the procedure of Algorithm 1 (CGSE), which is an elimination based method on the cumulative gain. As input, CGSE takes a set of arms $[k]$, and a confidence parameter $\delta$ (normally set to 0.1). Then, on each day, an active set of arms $\mathcal{A}$ is maintained and each is shown to users with equal probability $1/|\mathcal{A}|$. Finally, at the conclusion of each day, any arms that can be concluded to not have the maximum cumulative gain up through the current day among the active set are removed and never sampled again.

Formally, motivated by the existence of the always-valid confidence interval, CGSE eliminates an arm $j \in [k]$ on some day $t \geq 1$ when there exists an arm $i \in [k]$ such that $\widehat{G}_{i,t} - \widehat{G}_{i,j} - C(i, j, t, \delta/k) > 0$. This implies that the cumulative gain gap $G_{i,t} - G_{j,t} > 0$ is positive and thus arm $j \in [k]$ is suboptimal as judged by the cumulative gain metric with high probability. This procedure controls the variance of the cumulative gain estimator by keeping the sampling probabilities uniform across the set of active arms, which in turn results in sample efficient identification. Moreover, the elimination mechanism controls the regret by ceasing to give any traffic to arms that are provably sub-optimal.

An immediate criticism of this method is the concern that under non-stationarity, we may eliminate an arm early in the experiment that may perform better later on. However, we developed this strategy based on real-life data where even though the daily means of arms may move significantly, the differences between arms is relatively constant as described in Case Study 2 (Section 2.2). In Section 4.3, we discuss our method's strong theoretical guarantees.

# 4 EXPERIMENTS AND GUARANTEES

We now present an illustrative set of both offline and online production experiments. In our offline experiments, we highlight the benefits of our algorithm for identification and regret on a realistic example versus alternatives. The online experiments show that our algorithm has been deployed in production and delivered promising outcomes in comparison to standard regret minimization.

## 4.1 Offline Experiments

We consider a variation of a logged past online experiment and compare our method against several algorithmic baselines. Figure 3a plots the arm means on each day of the experiment. As arm 5 has the highest mean among the arms on each day, it is as the counterfactual optimal arm at any day. We run each of the candidate algorithms 100 times on this data using a daily batch size of 10000, and plot the mean regret of the algorithms over the runs (Figure 3b), the mean regret of the algorithms on the day the optimal arm is identified with statistical significance[5] (Figure 3c), and the probability over the runs of identifying the optimal arm with statistical significance by each given day (Figure 3d). Specifically, we monitor each algorithm using the always-valid confidence intervals from Section 3.2.3.

*Comparison Algorithms.* We consider CGSE and several comparisons. Thompson Sampling (TS) is an algorithm that maintains a posterior distribution on the running empirical mean of each arm that can be translated to a posterior probability $p_{i,t} = \alpha_{i,t}$ to play each

---

[5]If algorithm did not meet termination criteria, we use the algorithm's regret at the end of the final day.

arm $i \in [k]$ at day $t \geq 1$. Top Two TS (TTTS) [18] is a simple variant of TS intended for best arm identification that plays arm $i \in [k]$ at day $t \geq 1$ with probability $p_{i,t} = \alpha_{i,t}(\beta + (1 - \beta) \sum_{j \neq i} \frac{\alpha_{j,t}}{1 - \alpha_{j,t}})$ with $\beta = 1/2$. The uniform sampling algorithm plays each arm $i \in [k]$ at day $t \geq 1$ with probability $p_{i,t} = 1/k$. Finally, the best of both worlds (BOB) algorithm [1] sorts and ranks arms by decreasing order of the cumulative gain estimates and then computes a probability distribution based on this ranking. The probability of arm $i \in [k]$ being played at day $t \geq 1$ is given by $p_{i,t} = 1/(\widetilde{\langle i \rangle}_t \overline{\log}k)$ where $\widetilde{\langle i \rangle}_t$ denotes the cumulative gain estimate rank among the arms and $\overline{\log}k = \sum_{k'=1}^{k}(1/k')$.

*Experiment Results.* As expected, TS minimizes regret for this problem, but with high probability it is never able to end the experiment and identify the optimal arm. This is precisely because regret minimizing algorithms do not allocate enough impressions to suboptimal arms, compromising the statistical power. While TTTS outperforms TS in terms of identification time, it suffers in terms of regret since in the limit it only gives $\beta = 1/2$ of the impressions to the optimal arm. Moreover, in comparison to the other algorithms, it fails to identify the optimal arm fast. This is primarily because the probability of playing any arm not in the pair that appears closest to being optimal quickly tends to zero, which inflates the confidence intervals on these arms and hinders identification. The uniform algorithm suffers the maximum regret both as a function of the number of days and at experiment termination time. It is also suboptimal for identification since it does not allocate more impressions to arms that appear closer to being optimal. CGSE has nearly equal regret to TS at termination time and at any day, as expected. Furthermore, of all algorithms, it identifies the optimal arm the fastest. This experiment demonstrates the potential of CGSE for obtaining the best of three worlds. Finally, BOB suffers nearly double the regret at the termination time as CGSE, while also being slower to identify the optimal arm. This is due to BOB being more conservative and not eliminating arms to guard against fully adversarial problems, whereas our approach is more aggressive. However, as we observe in this non-stationary experiment, CGSE is robust enough to handle real-world data.

## 4.2 Online Experiments

We now present results from tests in a production environment. Given a set of content (the arms), a control group C and a treatment group T are dialed up with each receiving 50% of the traffic. In each experiment group, identical sets of content are scheduled. A TS implementation [15] allocates traffic among the content in the control group C, while CGSE allocates traffic among the content in the treatment group T. We dialed up dozens of experiments with this setup, but highlight particularly interesting ones that capture general outcomes and our learnings. In particular, we focus on the themes of robustness to non-stationarity, along with the practical benefits of using always-valid inference with adaptive traffic allocations, together forming the best of three worlds. More experiments are presented in Appendix E.

### 4.2.1 Theme 1: Robustness to Non-Stationarity. TS is ubiquitous in production bandit and experimentation systems. Yet, it is based on the running empirical mean estimator and the assumption of

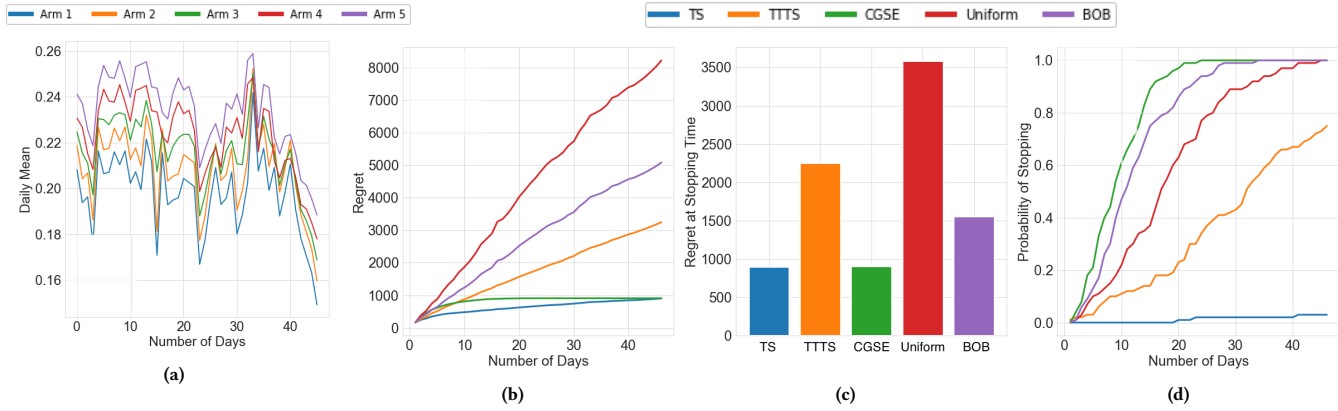

**Figure 3: Offline experiment 1. The daily arm means (a), regret as a function of the day (b), regret at the stopping time (c), and the probability of identifying the optimal arm with statistical significance by a given day (d).**

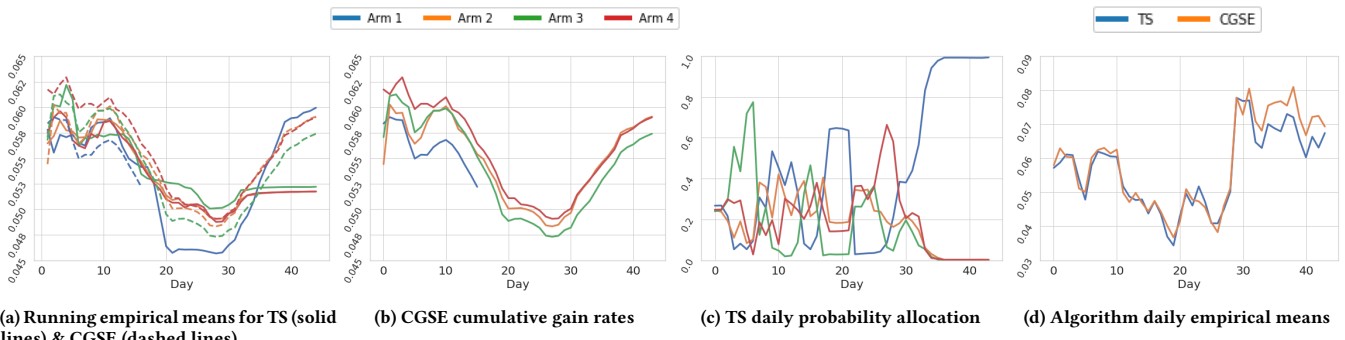

(a) Running empirical means for TS (solid lines) & CGSE (dashed lines)

(b) CGSE cumulative gain rates

(c) TS daily probability allocation

(d) Algorithm daily empirical means

**Figure 4: Live experiment 1: TS catastrophically fails on production data and shifts all traffic to the worst arm.**

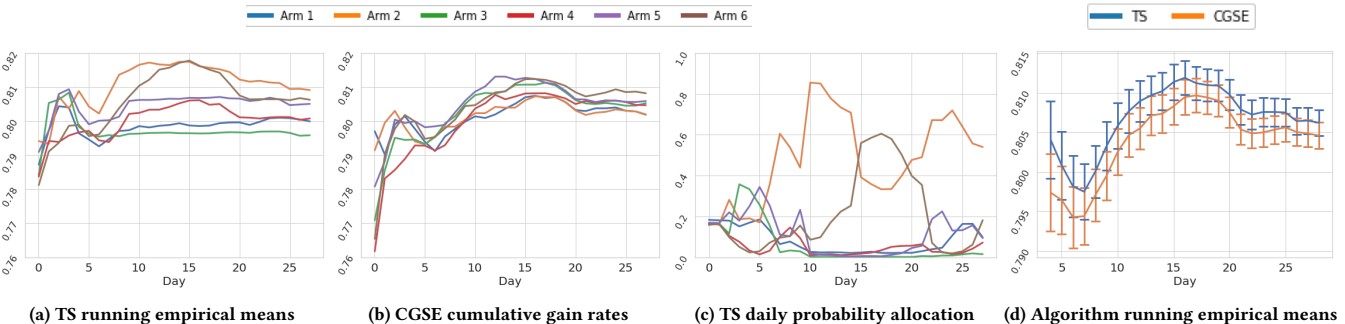

(a) TS running empirical means

(b) CGSE cumulative gain rates

(c) TS daily probability allocation

(d) Algorithm running empirical means

**Figure 5: Live experiment 2: TS fails to obtain significantly higher reward relative to CGSE and gives misleading inferences.**

a stochastic environment. Consequently, it lacks guarantees for drawing proper inferences or even minimizing regret on live traffic. Since CGSE acts akin to uniform sampling until arms are eliminated, the online experiments we conducted allowed us to effectively benchmark TS. The results show that TS fails to minimize regret in practice and it inflates decision-making error rates when using the collected data for inference. In contrast, our approach gives satisfying results with respect to the best of three worlds' objectives.

We illustrate these insights through the results of a pair of experiments with significant time-variation. Figure 4 shows the results of Experiment 1. After 2 weeks, CGSE eliminates Arm 1 (see Figure 4b), while TS was allocating nearly 100% of the traffic to this arm at the end of the experiment (see Figure 4c). This may appear to be an example where the arm switched from being the worst performer and became the best performer, but we can validate that this is not the case. In Figure 4a the solid lines represent the running empirical means of each arm as estimated from the TS algorithm, while the

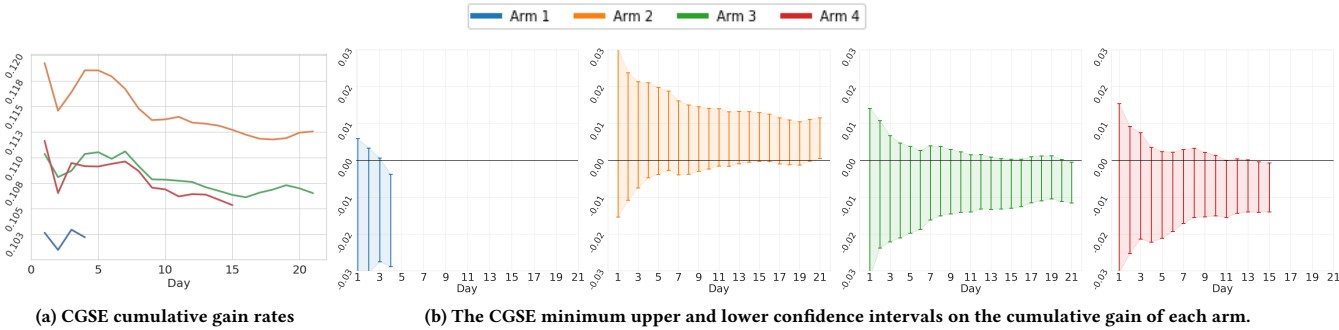

(a) CGSE cumulative gain rates

(b) The CGSE minimum upper and lower confidence intervals on the cumulative gain of each arm.

Figure 6: Live experiment 3: CGSE progressively eliminates suboptimal arms toward identifying the counterfactual optimal.

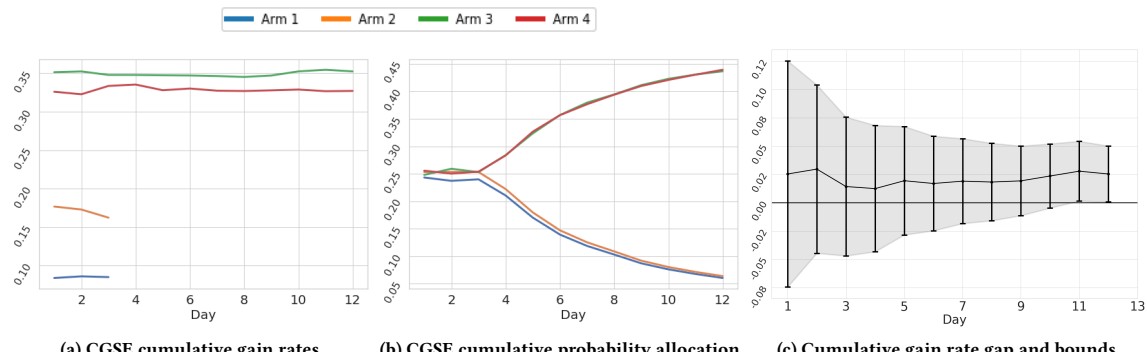

(a) CGSE cumulative gain rates

(b) CGSE cumulative probability allocation

(c) Cumulative gain rate gap and bounds

Figure 7: Live experiment 4: CGSE quickly eliminates highly suboptimal arms to minimize cost and increase power.

dashed lines represent the running empirical means of each arm as estimated from CGSE. We see that there is a huge bias downwards in the running empirical mean estimates of Arms 2-4 when comparing the data from TS (solid lines) and CGSE (dashed lines). This is due to the fact TS is giving little traffic to Arms 2-4 from day 30 onward as their underlying performance improves, while CGSE uniformly allocates traffic among Arms 2-4 and captures this effect. This suggests that TS's confidence in giving all of its traffic to Arm 1 is misplaced and instead CGSE was wise to eliminate Arm 1 early. Specifically, we see that the performance of all arms moves up as TS begins to switch its allocation to Arm 1 and this reinforcing feedback loop causes continued and exacerbated flawed allocations by TS. This is a real-world example of Simpson's paradox. As Figure 4d shows, this has an impact on the total reward obtained: the daily algorithm empirical means demonstrate that CGSE minimizes regret more effectively toward the end of the experiment.

Figure 5 shows the outcome of Experiment 2. The effects between arms in this experiment ended up being low relative to the amount of traffic. Consequently, CGSE did not eliminate any of the arms and simply produced a uniform traffic split over the course of the experiment. In contrast, TS again produces a highly dynamic traffic allocation and gives the vast majority of impressions to Arm 2 throughout much of the experiment (see Figure 5c). Yet, we observe that Arm 2 is in fact the arm with the lowest cumulative gain estimate using the data collected by CGSE (see Figure 5b), indicating that TS made a mistake in its allocation. This can also be observed

through Figure 5d, which shows that the running empirical means for the algorithms are nearly identical and statistically indistinguishable based on asymptotic confidence intervals. Moreover, standard hypothesis tests (t-test/z-test) on the running empirical mean using the data from TS (see Figure 5a) would have claimed that Arm 2 was positive significant relative to other arms midway through the experiment. This experiment highlights that TS is prone to costly decision-making mistakes (this is a potential Type 1 error) and fails to provide significant regret minimization benefits in production systems, while our approach overcomes the significant time-variation to provide valid inferences while minimizing the cost of experimentation.

*4.2.2 Theme 2: Efficient Always-Valid Inference and Finding the Best.* In most experiments, CGSE was able to identify the optimal arm within the usual time frames that an experiment is allowed to be active. We now discuss a pair of successful experiments that demonstrate the benefits of early elimination and the utility of always-valid confidence intervals for practical business uses.

In Experiment 3 (see Figure 6), the counterfactual optimal treatment was identified after 3 weeks of experimentation. By early elimination of suboptimal arms, CGSE was able to direct a higher percentage of traffic to the best performing arms and accelerate the comparison. We visualize the internal behavior of the algorithm in Figure 6b by plotting $\min_{j \in \mathcal{A}_t} \widehat{G}_{i,j,t} - C(i, j, t, \delta/k)$ and $\min_{j \in \mathcal{A}_t} \widehat{G}_{i,j,t} + C(i, j, t, \delta/k)$ (normalized to a rate) for each day $t$

that an arm $i \in [k]$ was active, where $\mathcal{A}_t$ denotes the day's set of active arms. These quantities respectively correspond to the lower and upper bounds of the minimum cumulative gain gap. They can be interpreted as the maximum potential loss (and gain, respectively) relative to the set of active arms, which holds with high probability by the always-valid confidence intervals.

The algorithm eliminates an arm when the minimum upper bound moves below zero, since this means that with high probability there exists another arm with higher cumulative gain. Moreover, the optimal arm has been identified when the lower bound moves above zero, since this means that with high probability the arm has the maximum cumulative gain among the active set. Beyond providing interpretation of the algorithm's behavior, these confidence intervals have the practical utility of being able to quantify the maximum potential loss or gain from making the decision to select an arm from an experiment. This is useful when business constraints necessitate early termination of an experiment.

Figure 7 shows the outcome of Experiment 4. In this experiment, Arms 1 and 2 had extremely low cumulative gain rates as compared to Arms 3 and 4. Consequently, CGSE eliminated them after just 3 days and split traffic evenly among Arms 3 and 4. By the end of the experiment when Arm 3 was identified as the counterfactual optimal, nearly 90% of the traffic had been allocated to Arms 3 and 4 (see Figure 7b where Arm 3 and 4 overlap). This experiment highlights the importance of adaptive experimentation. A naive uniform allocation would have been extremely costly as well as required a longer experiment, whereas early elimination limits the damage of introducing low-performing arms in an experiment and maximizes the decision-making power for identifying the optimal arm. This can be seen through the always-valid confidence intervals $\widehat{G}_{i,j,t} \pm C(i, j, t, \delta/k)$ (normalized to a rate) for each day $t$ on the gap between arm $i = 3$ and arm $j = 4$ shown in Figure 7c. As more data is collected by the algorithm, the confidence interval steadily shrinks until the lower bound moves above zero, at which point it concludes with high probability that Arm 3 is optimal.

## 4.3 Algorithmic Guarantees

This section shows that CGSE has theoretical guarantees for the objective of identifying the counterfactual optimal arm in experiments with daily time-variation, and that it seamlessly adapts to easier, stochastic experimentation data. Proofs are in Appendix D.

### 4.3.1 Guarantees for Correct Inference.
For this analysis, we restrict the allowable form of time-variation to make the fixed-confidence identification problem well-defined, but remark that it still captures a number of meaningful real-world scenarios. Toward this goal, let us define the cumulative gain rate gap relative to an arm $i^* \in [k]$ for any other arms $j \in [k]$ at any day $t \geq 1$ as the following:

$$\Delta_{j,t} := (\bar{n}_t)^{-1} \sum_{\tau=1}^{t} n_\tau (\mu_{i^*,\tau} - \mu_{j,\tau}).$$

ASSUMPTION 1. *There exists an arm $i^*$ such that for each arm $j \in [k] \setminus \{i^*\}$ the cumulative gain rate gap $\Delta_{j,t} \geq 0$ for all $t \geq 1$.*

Assumption 1 implies that the counterfactual optimal arm is time-independent, which means $i^* \in \arg\max_{i \in [k]} G_{i,t}$ for all $t \geq 1$. Yet, this is a mild restriction given that it allows for daily-time variation among all arms and does not require that the optimal counterfactual arm has the highest mean on each given day. Under this assumption, arm $i^*$ is not eliminated by CGSE with high probability.

PROPOSITION 2. *CGSE with $\delta \in (0, 1)$ does not eliminate the optimal arm $i^* \in [k]$ with probability at least $1 - \delta$ under Assumption 1.*

We prove this by showing the lower confidence bound on the cumulative gain gap between any arm $j \neq i^*$ and $i^*$ cannot fall below zero. The above does not guarantee that we necessarily find the best arm. To do so, we need an additional assumption.

ASSUMPTION 2. *For each arm $j \in [k] \setminus \{i^*\}$ there exists a day $t_0 > 0$ and $\epsilon > 0$ such that for $t > t_0$, $\Delta_{j,t} \geq \epsilon$.*

PROPOSITION 3. *CGSE with $\delta \in (0, 1)$ eliminates each arm $j \in [k] \setminus \{i^*\}$ with probability at least $1 - \delta$ under Assumptions 1–2.*

We prove this by arguing that the lower confidence bound on the cumulative gain rate gap between arm $i^*$ and any other arm $j \neq i^*$ must move above $\epsilon$, since the normalized confidence intervals necessarily shrink toward zero as a result of the allocation.

Put together, Propositions 2 and 3 allow us to conclude that, with probability at least $1 - \delta$, the optimal arm is returned by CGSE under Assumptions 1 and 2. This gives a strong guarantee in a general time-varying setting for the correctness of CGSE.

### 4.3.2 Guarantees in Stochastic Environments & Comparison.
In the stochastic stationary setting, CGSE reduces to a closely related version of the classical successive elimination algorithm [5]. Thus, it obtains the known guarantees for the algorithm in this situation. Specifically, if the environment is stationary, we have a guarantee that the algorithm will return the optimal arm with probability at least $1 - \delta$ in a number of samples not exceeding $O(\log(k/\delta) \sum_{i=1}^{k} \Delta_i^{-2})$ with regret at most $O(\log(k/\delta) \sum_{i=1}^{k} \Delta_i^{-1})$, both of which are near-optimal [5, 13].

In the best-arm identification literature there is little work about handling non-stochastic environments. Our work is perhaps most closely linked to the Best-of-Both-Worlds setting [1]. The authors propose a new sample complexity, $H_{BOB}$, which is necessarily larger than $\sum_{i=1}^{k} \Delta_i^{-2}$. They provide a conservative algorithm (see Section 4.1) which in the stochastic case requires no more samples than $H_{BOB} \log^2(k) \log(1/\delta)$ while being optimal in the adversarial case. We take a different, more aggressive approach that effectively guarantees an optimal sample complexity in stochastic settings, but give up on strong guarantees in fully adversarial settings.

## 5 CONCLUSION

This paper provides a rigorous discussion of experimentation objectives and the trade-offs of algorithmic methods for optimizing them. It demonstrates the shortfalls of the standard metric and estimator for inference when traffic is dynamically allocated and performance is time-varying. It proposes a metric, cumulative gain, that measures the counterfactual of the expected reward an arm could obtain if it received all the traffic. We propose an unbiased estimator of this metric, and empirically validate it. Finally, we combine cumulative gain estimators, always-valid confidence intervals, and an elimination algorithm to form a novel experimentation system that provides a robust and flexible tool for sequentially monitoring of experiments with fast identification guarantees at minimal cost.

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

## A PROOFS FOR SECTION 3 (OUR APPROACH)

PROPOSITION 1. *For any arm $i \in [k]$ and day horizon $T$, the estimator $\widehat{G}_{i,T} = \sum_{t=1}^{T} (r_{i,t}/p_{i,t})$ is unbiased for the cumulative gain. That is, we have $\mathbb{E}[\widehat{G}_{i,T}] = G_{i,T}$ as defined in Equation (2).*

PROOF. Assume that on each day $t \in [T]$ of the experiment, a probability vector $p_t = (p_{1,t}, \cdots, p_{k,t}) \in \Delta_k$ is chosen according to the history up to day $t$. Then, each visitor $s_t \in [n_t]$ on day $t \in [T]$ is shown an arm $I_{s_t} \in [k]$ that is selected with probability $\mathbb{P}(I_{s_t} = i) = p_{i,t}$ and a corresponding reward $r_{s_t}$ is observed. Formally, define the history up to any day $t \in [T]$ by the filtration

$$\mathcal{F}_{t-1} = \{\cup_{\tau=1}^{t-1} \cup_{s_t=1}^{n_\tau} (r_{s_t}, I_{s_t}, p_\tau)\}. \tag{6}$$

Now, to see that this is an unbiased estimator of the cumulative gain, observe that we have the following,

$$\mathbb{E}[\widehat{G}_{i,T}] = \mathbb{E}\Big[\sum_{t=1}^{T} \frac{r_{i,t}}{p_{i,t}}\Big]$$

$$= \sum_{t=1}^{T} \mathbb{E}\Big[\frac{1}{p_{i,t}} \sum_{s_t=1}^{n_t} \mathbb{1}\{I_{s_t} = i\} r_{s_t}\Big]$$

$$= \sum_{t=1}^{T} \mathbb{E}\Big[\frac{1}{p_{i,t}} \mathbb{E}\Big[\sum_{s_t=1}^{n_t} \mathbb{1}\{I_{s_t} = i\} r_{s_t} | \mathcal{F}_{t-1}\Big]\Big]$$

$$= \sum_{t=1}^{T} \mathbb{E}\Big[\frac{1}{p_{i,t}} \sum_{s_t=1}^{n_t} p_{i,t} \mu_{i,t}\Big]$$

$$= \sum_{t=1}^{T} n_t \mu_{i,t}.$$

This shows the estimator $\widehat{G}_{i,T}$ is unbiased since $\mathbb{E}[\widehat{G}_{i,T}] = G_{i,T}$. □

## B FIXED EFFECT VARIANCE REDUCED ESTIMATION

We now provide details on the variance reducing cumulative gain (VCGR) estimator described in Section 3.2.3 for situations in which the gap between any pair of arms is constant on each day. Specifically, we assume that there is a shift $\gamma_t$ on each day $t \in [T]$ of an experiment so that the mean of any arm $i \in [k]$ on day $t$ is given by $\mu_{i,t} = \mu_i + \gamma_t$. Consequently, the gap between a pair of arms $i, j \in [k]$ given by $\Delta := (\mu_i + \gamma_t) - (\mu_j + \gamma_t) = \mu_i - \mu_j$ and is constant on each day. In this situation, we are interested in measuring the gap $\Delta$, and our proposed Variance Reduced Cumulative Gain (VRCG) estimator is given by:

$$\widehat{G}_{i,j,T}^* = \sum_{t=1}^{T} (\widehat{\mu}_{i,t} - \widehat{\mu}_{j,t}) w_t,$$

where

$$w_t = \big(\sum_{t=1}^{T} (n_{i,t}^{-1} + n_{j,t}^{-1})^{-1}\big)^{-1} (n_{i,t}^{-1} + n_{j,t}^{-1})^{-1} \quad \forall \, t \in [T].$$

For the analysis to follow, we assume that the arm sample counts on day $t \in [T]$ are non-zero and independent of the history up to day $t$ defined by the filtration in Equation (6). As a result of this independence, the variance reduced cumulative gain estimator is an unbiased estimate of the difference of means for any arms $i, j \in [k]$ after $T$ days of the experiment:

$$\mathbb{E}\Big[\sum_{t=1}^{T} (\widehat{\mu}_{i,t} - \widehat{\mu}_{j,t}) w_t\Big] = \sum_{t=1}^{T} \mathbb{E}[(\widehat{\mu}_{i,t} - \widehat{\mu}_{j,t})] w_t = \sum_{t=1}^{T} (\mu_i - \mu_j) w_t = \mu_i - \mu_j.$$

In the following result and proof, we show that the variance reduced cumulative gain estimator minimizes the variance of the estimate.

PROPOSITION 4. *The variance of the estimator $\sum_{t=1}^{T} (\widehat{\mu}_{i,t} - \widehat{\mu}_{j,t}) w_t$ is minimized with $w_t = (1/n_{i,t} + 1/n_{j,t})^{-1} \big(\sum_{\tau=1}^{T} (1/n_{i,\tau} + 1/n_{j,\tau})^{-1}\big)^{-1}$ for all $t \in [T]$ and the resulting variance of the estimator is $\big(\sum_{\tau=1}^{T} (1/n_{i,\tau} + 1/n_{j,\tau})^{-1}\big)^{-1}$.*

PROOF. We formulate the following optimization problem to select a vector of weights $w \in \mathbb{R}^T$ constrained to the simplex that minimizes the variance of the estimator as follows:

$$\min_{w \in \mathbb{R}^T} \quad \mathbb{V}\Big[\sum_{t=1}^{T} (\widehat{\mu}_{i,t} - \widehat{\mu}_{j,t}) w_t\Big]$$

$$\text{such that} \quad w \geq 0 \text{ and } \sum_{t=1}^{T} w_t = 1.$$

This problem can be rewritten as follows after denoting the variance of $\widehat{\mu}_{i,t} - \widehat{\mu}_{j,t}$ as $\sigma_t^2$ for each $t \in [T]$:

$$\min_{w \in \mathbb{R}^T} \quad \sum_{t=1}^{T} w_t^2 \sigma_t^2$$

$$\text{such that} \quad w \geq 0 \text{ and } \sum_{t=1}^{T} w_t = 1.$$

To solve this optimization problem, we write out the Lagrangian and the associated KKT conditions. In particular, we have the following Lagrangian:

$$\mathcal{L}(w, \lambda, \gamma) = \sum_{t=1}^{T} w_t^2 \sigma_t^2 + \lambda \left( \sum_{t=1}^{T} w_t - 1 \right) + \sum_{t=1}^{T} \gamma_t (-w_t).$$

The resulting KKT conditions are given by:

$$\forall\, t = 1, 2, \ldots, T : \; 2 w_t \sigma_t^2 + \lambda - \gamma_t = 0 \quad \text{(stationarity)}$$

$$\forall\, t = 1, 2, \ldots, T : \; \gamma_t (-w_t) = 0 \quad \text{(complementary slackness)}$$

$$\sum_{t=1}^{T} w_t = 1, w \geq 0 \quad \text{(primal feasibility)}$$

$$\gamma \geq 0 \quad \text{(dual feasibility)}.$$

Observe that this simplifies to the following conditions by taking $\gamma = 0$ and absorbing constants into the multiplier $\lambda$:

$$\forall\, t = 1, 2, \ldots, T : \; w_t \sigma_t^2 = \lambda \quad \text{(stationarity)}$$

$$\sum_{t=1}^{T} w_t = 1, w \geq 0 \quad \text{(primal feasibility)}.$$

Now, since we require $w_t \sigma_t^2 = w_T \sigma_T^2 = \lambda$ for each $t \in [T-1]$, the conditions can equivalently be written as:

$$\forall\, t = 1, 2, \ldots, T-1 : \; w_t = w_T \sigma_T^2 / \sigma_t^2 \quad \text{(stationarity)}$$

$$\sum_{t=1}^{T} w_t = 1, w \geq 0 \quad \text{(primal feasibility)}.$$

To find the solution, we reformulate the conditions into the simplified set given below:

$$\sum_{t=1}^{T} \underbrace{w_T \sigma_T^2 / \sigma_t^2}_{w_t} = 1 \quad \text{and} \quad w \geq 0.$$

It now becomes clear by solving for $w_T$ in the equation above and applying the constraint $w_t = w_T \sigma_T^2 / \sigma_t^2$ for all $t \in [T-1]$ the solution is

$$w_T = \frac{1}{\sigma_T^2 \sum_{t=1}^{T} \sigma_t^{-2}} \quad \text{and} \quad w_t = w_T \sigma_T^2 / \sigma_t^2 \; \forall\, t \in [T-1].$$

The solution that satisfies these conditions is given by

$$w_t = \frac{\sigma_T^2}{\sigma_t^2} \frac{1}{\sigma_T^2 \sum_{\tau=1}^{T} \sigma_\tau^{-2}} = \frac{1}{\sigma_t^2 \sum_{\tau=1}^{T} \sigma_\tau^{-2}} = \frac{\sigma_t^{-2}}{\sum_{\tau=1}^{T} \sigma_\tau^{-2}}.$$

It then follows from plugging this quantity into the variance definition for the estimator that for this choice of $w$ we have

$$\mathbb{V}\left[ \sum_{t=1}^{T} (\widehat{\mu}_{i,t} - \widehat{\mu}_{j,t}) w_t \right] = \sum_{t=1}^{T} w_t^2 \sigma_t^2 = \sum_{t=1}^{T} \frac{\sigma_t^{-2}}{\left( \sum_{\tau=1}^{T} \sigma_\tau^{-2} \right)^2} = \frac{1}{\sum_{\tau=1}^{T} \sigma_\tau^{-2}}.$$

Thus, with $\sigma_t^2 = (\mu_{i,t}(1 - \mu_{i,t}) / n_{i,t} + \mu_{j,t}(1 - \mu_{j,t}) / n_{j,t})$, we have

$$w_t = \frac{(\mu_{i,t}(1 - \mu_{i,t}) / n_{i,t} + \mu_{j,t}(1 - \mu_{j,t}) / n_{j,t})^{-1}}{\sum_{\tau=1}^{T} (\mu_{i,\tau}(1 - \mu_{i,\tau}) / n_{i,\tau} + \mu_{j,\tau}(1 - \mu_{j,\tau}) / n_{j,\tau})^{-1}},$$

and

$$\mathbb{V}\left[ \sum_{t=1}^{T} (\widehat{\mu}_{i,t} - \widehat{\mu}_{j,t}) w_t \right] = \frac{1}{\sum_{\tau=1}^{T} (\mu_{i,\tau}(1 - \mu_{i,\tau}) / n_{i,\tau} + \mu_{j,\tau}(1 - \mu_{j,\tau}) / n_{j,\tau})^{-1}}.$$

In practice, the daily means would be unknown. Thus, we consider minimizing an upper bound on the variance of the estimator by defining $\sigma_t^2 = (1/n_{i,t} + 1/n_{j,t})/4$ using that $x(1-x) \leq 1/4$ for $x \in (0, 1)$ and the analysis follows identically with

$$w_t = \frac{(1/(4 n_{i,t}) + 1/(4 n_{j,t}))^{-1}}{\sum_{\tau=1}^{T} (1/(4 n_{i,\tau}) + 1/(4 n_{j,\tau}))^{-1}} = \frac{(1/n_{i,t} + 1/n_{j,t})^{-1}}{\sum_{\tau=1}^{T} (1/n_{i,\tau} + 1/n_{j,\tau})^{-1}}.$$

This gives a variance upper bound for the estimator of

$$\mathbb{V}\left[ \sum_{t=1}^{T} (\widehat{\mu}_{i,t} - \widehat{\mu}_{j,t}) w_t \right] \leq \frac{1}{\sum_{\tau=1}^{T} (1/n_{i,\tau} + 1/n_{j,\tau})^{-1}}.$$

□

# C  VARIANCE ANALYSIS AND ALWAYS-VALID CONFIDENCE INTERVAL DERIVATION

We now provide an analysis comparing the variance of the running empirical mean estimator and the cumulative gain estimator and go into more details describing the derivation of the always-valid confidence intervals adopted on cumulative gain differences.

## C.1  Variance Analysis of Running Empirical Mean and Cumulative Gain Estimators

The purpose of this analysis is to compare how the variance of the running empirical mean and cumulative gain estimators behave in stationary environments where they are both unbiased, as well as to gain insights into how the variance of the cumulative gain estimator can be controlled through the allocation of samples among arms. In this analysis, we focus on a stochastic environment in which for an arm $i \in [k]$ the underlying daily mean $\mu_{i,t} = \mu_i$ is fixed for all $t \in [T]$. Moreover, we work under the assumptions described when formulating and analyzing the cumulative gain estimator in Section 3.2. For the sake of simplifying the discussion and avoiding issues around considering conditional expectations, we also assume that the daily probability allocation $p_{i,t}$ for the arm is fixed ahead of the experiment for all $t \in [T]$.

In this analysis, we consider another version of the standard running empirical mean estimator since the definition from Equation (1) is not necessarily unbiased under the assumptions as a result of randomness in the observed sample count, and so that the underlying metric being estimated has equivalent units to the cumulative gain for the sake of comparison. In particular, let us define the estimator

$$\bar{\mu}'_{i,T} := \sum_{\tau=1}^{T} \frac{\sum_{t=1}^{T} r_{i,t}}{\sum_{t=1}^{T} p_{i,t}}.$$

Following an argument analogous to the proof of Proposition 1, it can be shown that $\bar{\mu}'_{i,T}$ is an unbiased estimate of the mean $\mu_i$ when normalized by the total experiment sample count $\sum_{t=1}^{T} n_t$.

We now calculate the variance $\mathbb{V}(\widehat{G}_{i,T})$ of the cumulative gain estimator and the variance $\mathbb{V}(\bar{\mu}'_{i,T})$ of the running empirical mean estimator. To begin, observe that on any day $t \in [T]$ and observation $s_t \in [n_t]$, we have the following bound:

$$\begin{aligned}
\mathbb{V}(\mathbb{1}\{I_{s_t} = i\}r_{s_t}) &= \mathbb{E}[\mathbb{V}(\mathbb{1}\{I_{s_t} = i\}r_{s_t}|I_{s_t})] + \mathbb{V}(\mathbb{E}[\mathbb{1}\{I_{s_t} = i\}r_{s_t}|I_{s_t}]) \\
&= \mathbb{E}[\mathbb{1}\{I_{s_t} = i\}\mu_{I_{s_t}}(1 - \mu_{I_{s_t}})] + \mathbb{V}(\mathbb{1}\{I_{s_t} = i\}\mu_{I_{s_t}}) \\
&= p_{i,t}\mu_i(1 - \mu_i) + p_{i,t}(1 - p_{i,t})\mu_i^2 \\
&= p_{i,t}\mu_i - p_{i,t}^2\mu_i^2 \\
&\leq p_{i,t}\mu_i.
\end{aligned}$$

Given this calculation, we obtain the variance of the running empirical mean estimator as follows:

$$\begin{aligned}
\mathbb{V}(\bar{\mu}'_{i,T}) &= \frac{1}{(\sum_{t=1}^{T} p_{i,t})^2} \mathbb{V}\left( \sum_{\tau=1}^{T} \sum_{t=1}^{T} \sum_{s_t=1}^{n_t} \mathbb{1}\{I_{s_t} = i\}r_{s_t} \right) \\
&\leq \frac{T^2}{(\sum_{t=1}^{T} p_{i,t})^2} \sum_{t=1}^{T} n_t p_{i,t}\mu_i.
\end{aligned}$$

Similarly, the variance of the cumulative gain estimator is obtained as follows:

$$\begin{aligned}
\mathbb{V}(\widehat{G}_{i,T}) &= \mathbb{V}\left( \sum_{t=1}^{T} \frac{1}{p_{i,t}} \sum_{s=1}^{n_t} \mathbb{1}\{I_{s_t} = i\}r_{s_t} \right) \\
&\leq \sum_{t=1}^{T} \frac{n_t \mu_i}{p_{i,t}}.
\end{aligned}$$

To facilitate the discussion to follow, let us now assume that the daily experiment traffic $n_t = n$ is fixed for each day $t \in [T]$ of the experiment. Then, summarizing, we have shown

$$\mathbb{V}(\bar{\mu}'_{i,T}) \leq n\mu_i \frac{T^2}{\sum_{t=1}^{T} p_{i,t}} \quad \text{and} \quad \mathbb{V}(\widehat{G}_{i,T}) \leq n\mu_i \sum_{t=1}^{T} \frac{1}{p_{i,t}}. \tag{7}$$

*Implications.* This analysis shows that in general, if the underlying environment is indeed stationary, the variance of the cumulative gain estimator $\widehat{G}_{i,T}$ can be higher than that of the running empirical mean estimator $\bar{\mu}'_{i,T}$. This follows from the arithmetic-mean-harmonic-mean inequality, which in this context implies

$$\frac{T^2}{\sum_{t=1}^{T} p_{i,t}} \leq \sum_{t=1}^{T} \frac{1}{p_{i,t}} \quad \text{so that} \quad \mathbb{V}(\bar{\mu}'_{i,T}) \leq \mathbb{V}(\widehat{G}_{i,T}).$$

Observe that given a static probability allocation across time as in standard A/B/N testing, we have $\mathbb{V}(\bar{\mu}'_{i,T}) = \mathbb{V}(\widehat{G}_{i,T})$. In contrast, the variance of the cumulative gain estimator can be much greater than the variance of the running empirical mean estimator if the probability allocation is highly dynamic or tends toward zero on any given day. We remark that the variance analysis for the cumulative gain estimator

follows similarly without the stationarity assumption used in this context. Consequently, Algorithm 1 presented in Section 3.3 for adaptive counterfactual inference is motivated by controlling the variance of the estimator through the allocation.

## C.2 Always-Valid Confidence Interval Justification

We construct an always-valid confidence interval based on normal approximations and an application of the mixture sequential probability ratio test (MSPRT) [8, 12, 17]. Here, we walk through the derivation of the always-valid confidence interval we adopt. This begins by examining the variance process of the cumulative gain gap estimator $\widehat{G}_{i,j,t} := \widehat{G}_{i,t} - \widehat{G}_{j,t}$ at any arbitrary day $t \in [T]$. In general, conditional on $\mathcal{F}_{t-1}$ (see Eq. 6),

$$r_{i,t} \sim \text{Binomial}(n_{i,t}, \mu_{i,t}) \quad \text{and} \quad r_{j,t} \sim \text{Binomial}(n_{j,t}, \mu_{j,t}).$$

Then, assuming the total daily sample count $n_t$ is sufficiently large, observing that $\mathbb{E}[n_{i,t}] = n_t p_{i,t}$ and $\mathbb{E}[n_{j,t}] = n_t p_{j,t}$, and invoking the central limit theorem, we have

$$r_{i,t} \sim \mathcal{N}(n_t p_{i,t} \mu_{i,t}, n_t p_{i,t} \mu_{i,t}(1 - \mu_{i,t})) \quad \text{and} \quad r_{j,t} \sim \mathcal{N}(n_t p_{j,t} \mu_{j,t}, n_t p_{j,t} \mu_{j,t}(1 - \mu_{j,t})).$$

Thus, we see that conditionally,

$$\frac{r_{i,t}}{p_{i,t}} - \frac{r_{j,t}}{p_{j,t}} \sim \mathcal{N}\Big(n_t(\mu_{i,t} - \mu_{j,t}), n_t\Big(\frac{\mu_{i,t}(1 - \mu_{i,t})}{p_{i,t}} + \frac{\mu_{j,t}(1 - \mu_{j,t})}{p_{j,t}}\Big)\Big)$$

Now, using this approximation, we define $S_t = \widehat{G}_{i,j,t} - G_{i,j,t}$ and the corresponding variance process

$$V_t = \sum_{\tau=1}^{t} n_\tau \Big(\frac{\mu_{i,\tau}(1 - \mu_{i,\tau})}{p_{i,\tau}} + \frac{\mu_{j,\tau}(1 - \mu_{j,\tau})}{p_{j,\tau}}\Big).$$

Finally, under the distributional assumptions and applications of the central limit theorem, we see that $\{S_t\}_{t \geq 1}$ and the process $\{V_t\}_{t \geq 1}$ are adapted to $\mathcal{F}_t$ and satisfy the property that the process defined by $\{\exp(S_t - \lambda^2/(2V_t))\}_{t \geq 1}$ forms a supermartingale for any $\lambda \geq 0$.

To obtain the always-valid confidence interval, we now apply the MSPRT (see Eq. 14 in [8]) using the plugin estimators $\widehat{\mu}_{i,t}$ and $\widehat{\mu}_{j,t}$ for the unknown daily arm means $\mu_{i,t}$ and $\mu_{j,t}$ on each day in an estimate of the variance. As a result, under the stated assumptions, we have

$$\mathbb{P}(\exists\, t \geq 1, i, j \in [k] : |\widehat{G}_{i,j,t} - G_{i,j,t}| \geq C(i, j, t, \delta)) \leq \delta,$$

with

$$C(i, j, t, \delta) := \sqrt{(\widehat{V}_{i,j,t} + \rho) \log\big((\widehat{V}_{i,j,t} + \rho)/(\rho \delta^2)\big)}$$

where $\rho > 0$ is a fixed constant and

$$\widehat{V}_{i,j,t} = \sum_{\tau=1}^{t} n_\tau \Big(\frac{\widehat{\mu}_{i,\tau}(1 - \widehat{\mu}_{i,\tau})}{p_{i,\tau}} + \frac{\widehat{\mu}_{j,\tau}(1 - \widehat{\mu}_{j,\tau})}{p_{j,\tau}}\Big).$$

## D PROOFS FOR SECTION 4 (EXPERIMENTS AND GUARANTEES)

PROPOSITION 2. *CGSE with $\delta \in (0, 1)$ does not eliminate the optimal arm $i^* \in [k]$ with probability at least $1 - \delta$ under Assumption 1.*

PROOF. Let us begin by defining the 'good event', in which the always-valid confidence intervals on the cumulative gain gap relative to the optimal arm hold simultaneously. That is,

$$\mathcal{E} = \{\forall\, t \geq 1, \forall\, j \in [k] : \big|\widehat{G}_{i^*,j,t} - G_{i^*,j,t}\big| \leq C(i^*, j, t, \delta/k)\}.$$

Note that this event holds with probability at least $1 - \delta$ by the definition of the always-valid confidence interval and a union bound.

We claim that the optimal arm $i^* \in [k]$ is not eliminated by CGSE on the event $\mathcal{E}$ holding. This will allow us to conclude that the optimal arm $i^* \in [k]$ is not eliminated by CGSE with probability at least $1 - \delta$. Define $\bar{n}_t = \sum_{\tau=1}^{t} n_\tau$ for any day $t \geq 1$. Then, for any arm $j \in [k] \setminus \{i^*\}$ on any day $t \geq 1$, on the event $\mathcal{E}$, we have the following which is explained below:

$$\bar{n}_t^{-1}\Big(\widehat{G}_{j,i^*,t} - C(j, i^*, t, \delta/k)\Big) = \bar{n}_t^{-1}\Big(\mathbb{E}[\widehat{G}_{j,i^*,t}] + \widehat{G}_{j,i,t} - \mathbb{E}[\widehat{G}_{j,i^*,t}] - C(j, i^*, t, \delta/k)\Big) \tag{8}$$

$$= -\Delta_{j,t} + \bar{n}_t^{-1}\Big(\widehat{G}_{j,i^*,t} - \mathbb{E}[\widehat{G}_{j,i^*,t}] - C(j, i^*, t, \delta/k)\Big) \tag{9}$$

$$\leq -\Delta_{j,t} + \bar{n}_t^{-1}\Big(C(j, i^*, t, \delta/k) - C(j, i^*, t, \delta/k)\Big) \tag{10}$$

$$= -\Delta_{j,t} \tag{11}$$

$$\leq 0. \tag{12}$$

Observe that (8) is obtained by adding and subtracting $\mathbb{E}[\widehat{G}_{j,i^*,t}]$. Equation (9) uses the unbiased estimator property of the cumulative gain estimator from Proposition 1 so that $\mathbb{E}[\widehat{G}_{j,i^*,t}] = G_{j,i^*,t}$ and then applies the definition $\bar{n}_t^{-1} G_{j,i^*,t} = -\Delta_{j,t}$. Note that Equation (10) holds by the confidence bounds on the event $\mathcal{E}$. Moreover, Equation (11) is a direct simplification since $C(j, i^*, t, \delta/k) = C(i^*, j, t, \delta/k)$ by definition.

Finally, Equation (12) holds by Assumption 1. Thus, by the elimination criterion of CGSE, the optimal arm $i^* \in [k]$ is not eliminated on the event $\mathcal{E}$. Hence, with probability at least $1 - \delta$, the optimal arm is not eliminated. □

PROPOSITION 3. *CGSE with $\delta \in (0, 1)$ eliminates each arm $j \in [k] \setminus \{i^*\}$ with probability at least $1 - \delta$ under Assumptions 1–2.*

PROOF. Let us begin by defining the 'good event', in which the always-valid confidence intervals on the cumulative gain gap relative to the optimal arm hold simultaneously. That is,

$$\mathcal{E} = \{\forall\, t \geq 1, \forall\, j \in [k] : \left|\widehat{G}_{i^*,j,t} - G_{i^*,j,t}\right| \leq C(i^*, j, t, \delta/k)\}.$$

We claim that all suboptimal arms $j \in [k] \setminus \{i^*\}$ are eventually eliminated by CGSE on the event $\mathcal{E}$ holding. This will allow us to conclude that all suboptimal arms $j \in [k] \setminus \{i^*\}$ are eventually eliminated by CGSE with probability at least $1 - \delta$. Define $\bar{n}_t = \sum_{\tau=1}^{t} n_\tau$ for any day $t \geq 1$. Now, consider an arbitrary day $t \geq 1$ and consider some arm $j \in \mathcal{A}_t \setminus \{i^*\}$ that has not been eliminated yet and let $i = \arg\max_{i' \in \mathcal{A}_t \setminus \{j\}} \widehat{G}_{i,t}$ denote the active arm with the maximum cumulative gain estimate excluding arm $j$. Recall that by Proposition 2 and its proof, the optimal arm $i^* \in [k]$ always remains in the active set on the event $\mathcal{E}$. Then, on the event $\mathcal{E}$, we have the following that is explained below:

$$\bar{n}_t^{-1}\left(\widehat{G}_{i,j,t} - C(i, j, t, \delta/k)\right) \geq \bar{n}_t^{-1}\left(\widehat{G}_{i^*,j,t} - C(i, j, t, \delta/k)\right) \tag{13}$$

$$= \bar{n}_t^{-1}\left(\mathbb{E}[\widehat{G}_{i^*,j,t}] + \widehat{G}_{i^*,j,t} - \mathbb{E}[\widehat{G}_{i^*,j,t}] - C(i, j, t, \delta/k)\right) \tag{14}$$

$$\geq \bar{n}_t^{-1}\left(\mathbb{E}[\widehat{G}_{i^*,j,t}] - C(i^*, j, t, \delta/k) + C(i, j, t, \delta/k)\right) \tag{15}$$

$$= \Delta_{j,t} - \bar{n}_t^{-1}\left(C(i^*, j, t, \delta/k) + C(i, j, t, \delta/k)\right). \tag{16}$$

Observe that Equation (13) holds because $\widehat{G}_{i,j,t} \geq \widehat{G}_{i^*,j,t}$ since $i = \arg\max_{i' \in \mathcal{A}_t \setminus \{j\}} \widehat{G}_{i,j,t}$. Then, Equation (14) is obtained by adding and subtracting $\mathbb{E}[\widehat{G}_{i^*,j,t}]$ and Equation (15) holds by the confidence bounds on the event $\mathcal{E}$. Moreover, Equation (16) uses the unbiased estimator property of the cumulative gain estimator from Proposition 1 so that $\mathbb{E}[\widehat{G}_{i^*,j,t}] = G_{i^*,j,t}$ and then applies the definition $\bar{n}_t^{-1} G_{i^*,j,t} = \Delta_{j,t}$.

Now, by Assumption 2, there exists a day $t_0$ and constant $\epsilon > 0$ such that for day $t > t_0$, $\Delta_{j,t} \geq \epsilon$. Thus, continuing on, and considering $t > t_0$, we get the following:

$$\bar{n}_t^{-1}\left(\widehat{G}_{i,j,t} - C(i, j, t, \delta/k)\right) \geq \Delta_{j,t} - \bar{n}_t^{-1}\left(C(i^*, j, t, \delta/k) + C(i, j, t, \delta/k)\right)$$

$$\geq \epsilon - \bar{n}_t^{-1}\left(C(i^*, j, t, \delta/k) + C(i, j, t, \delta/k)\right).$$

We can conclude that the arm $j$ is eventually eliminated by CGSE as a result of the elimination criterion given the event $\mathcal{E}$ since $\bar{n}_t^{-1}\left(C(i^*, j, t, \delta/k) + C(i, j, t, \delta/k)\right) \to 0$ as $t \to \infty$. This follows from the observation that the confidence intervals grow asymptotically as $O(\sqrt{\bar{n}_t \log(\bar{n}_t)})$ owing to the sampling allocation, so the confidence intervals normalized by $\bar{n}_t^{-1}$ tend to zero as $t \to \infty$. Hence, with probability at least $1 - \delta$, CGSE eliminates each arm $j \in [k] \setminus \{i^*\}$ under the stated assumptions. □

# E SUPPLEMENTAL EXPERIMENTS

In this appendix, we present more online experiments. We begin with an experiment that illustrates many of the claims we have made in this paper. Following this, we mirror the structure of the experiments section in the main paper and present examples highlighting robustness to non-stationarity and efficient always-valid inference and finding the best. As described previously, in these experiments a control group C and a treatment group T are dialed up with each receiving 50% of the traffic. In each experiment group, identical sets of content are scheduled. TS allocates traffic among the content in the control group C, while CGSE allocates traffic among the content in the treatment group T.

## E.1 Experiment Case Study

In this case study, we go through full detail of an online experiment from which a number of illuminating observations can be made.

*Comparison of Allocations and Estimates.* We begin by presenting the daily traffic allocations and the resulting arm performance estimates to qualitatively compare the behaviors of the algorithms. In particular, the daily traffic allocations over the arm set are shown in Figures 8a–8b, while the observed arm performance estimates are shown in Figures 8c–8d. Observe that the TS algorithm produces a highly dynamic traffic allocation in an effort to maximize the accrued successes during the experiment (Figure 8a). In contrast, CGSE has a uniform traffic allocation over the arm set to begin and then arm are eliminated over time as soon as they can be proven to be suboptimal (Figure 8b). Together with the arm performance estimates resulting from the traffic allocations, a number of illuminating observations can be made from this data. Arm 2 and Arm 3 have the highest cumulative gain rates throughout the experiment in this data and nearly equal performance (see Figure 8d). Yet, the running empirical means resulting from the TS data are such that Arm 2 has a significant gap to Arm 3 and it even falls behind Arm 1 for a period of the experiment (see Figure 8c). This highlights a key problem with using the running empirical mean in combination with regret-minimizing algorithms for experimentation: performance estimates are biased and this can result in erroneous decision-making [20].

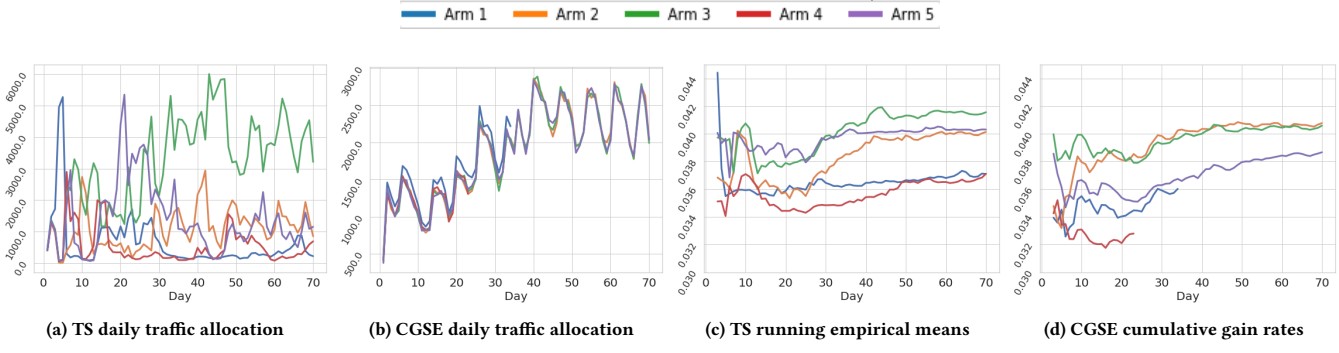

(a) TS daily traffic allocation     (b) CGSE daily traffic allocation     (c) TS running empirical means     (d) CGSE cumulative gain rates

Figure 8: Experiment case study traffic allocations and estimates.

Moreover, there are periods of the experiment in which the TS algorithm allocates the vast majority of traffic to suboptimal arms including Arm 1 and Arm 5 (Figure 8a). This highlights that heuristic decision-making rules based around the traffic allocation or the model posterior distribution of TS commonly result in flawed decision-making in experimentation, a fact that is known in the academic community but often brushed off in industry settings [12].

*Comparison of Confidence Intervals on Cumulative Gain.* So far, the results presented from this experiment have reinforced that accurate decision-making in experimentation cannot be made reliably from sample observations of adaptively collected data. Recall that our solution to combat this challenge is to base performance evaluations the cumulative gain metric. To quantify the uncertainty in these estimates and make comparisons between the performance of treatments, we rely on always-valid confidence intervals that can be monitored throughout an experiment. Crucially, the amount of uncertainty and consequently the size of the confidence intervals is highly dependent on the traffic allocation. Algorithms that balance traffic intelligently over treatments to reduce this uncertainty reach decisions faster, while methods that more greedily assign traffic to treatments in order to minimize experimentation cost take longer.

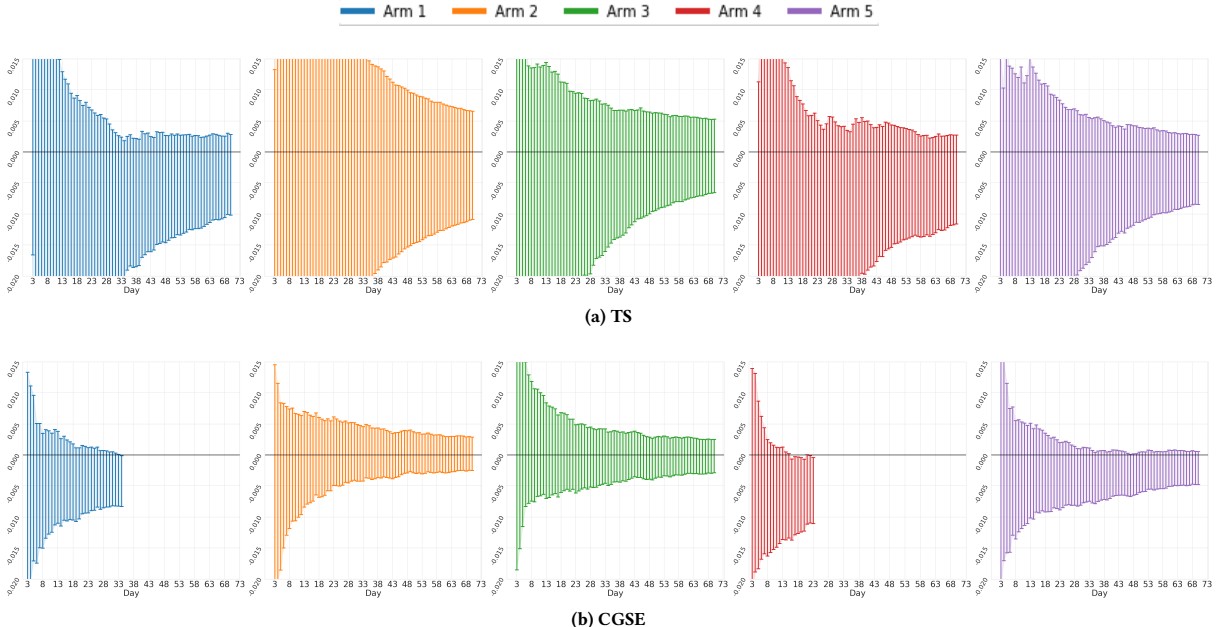

(a) TS

(b) CGSE

Figure 9: The minimum upper and lower confidence intervals on the cumulative gain of each arm.

We validate this through the experimental data presented in Figure 9. In particular, we show the minimum lower and upper bounds using the always-valid confidence interval on the cumulative gain rate for each arm in comparison to the active set of arms. That is, we show

$\min_{j \in \mathcal{A}_t} \widehat{G}_{i,j,t} - C(i,j,t,\delta/k)$ and $\min_{j \in \mathcal{A}_t} \widehat{G}_{i,j,t} + C(i,j,t,\delta/k)$ (normalized to a rate) for each day $t$ that an arm $i \in [k]$ was active, where $\mathcal{A}_t$ denotes the day's set of active arms. Again, these quantities can be interpreted as the maximum potential loss (and gain, respectively) relative to the set of active arms, which holds with high probability by the always-valid confidence intervals. The data resulting from TS and CGSE are presented in Figure 9a and Figure 9b, respectively. If the upper confidence bound is below zero, this indicates that the arm is not the counterfactual optimal. Observe that with the data from TS (Figure 9a), no definitive decisions can be made comparing the performance of arms. This is a direct result of the erratic traffic allocation. In contrast, the CGSE data (see Figure 9b) finds that both Arm 4 and Arm 1 are not optimal early in the experiment. Consequently, they were eliminated from consideration and the resulting traffic was shifted to the remaining arm for the rest of the experiment to increase decision-making power. Moreover, Arm 5 was close to being eliminated and likely would have been removed if the experiment continued on marginally longer. This data is evidence of the significant increase in decision-making power that results from CGSE and points to the potential for faster experiments and higher overall experimentation throughput by using the algorithm in place of TS.

*Comparison of Algorithm Regret Performance.* To finish our analysis of this representative experiment, we compare the running empirical means of the algorithms during the experiment. As has been alluded to, TS is an optimal procedure for minimizing regret in stationary environments. CGSE gives up some cost minimization in near stochastic environments to be able to make decisions faster. The trade-off is not symmetric between regret minimization and identification time, and consequently we expect that the loss in metric accrual during experiments is compensated for by the ability to make decisions faster and launch more experiments as a result. Figure 10 shows the running empirical means over the course of the experiment for TS and CGSE. As can be expected given that this experiment was only mildly non-stationary, TS outperforms CGSE by this metric. However, the gap is relatively small with a relative lift of 2.45% over the duration of the experiment. This can be the price of adaptive experimentation for inference, which is markedly less than the price of a standard A/B/N experiment.

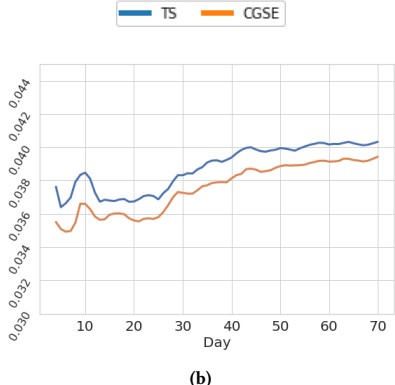

(b)

**Figure 10: Running Empirical means for TS and CGSE.**

## E.2 Robustness to Non-Stationarity and Always-Valid Inference

We now show more live experiment results that reinforce the observations made in the main body of the paper. Specifically, the robustness to non-stationarity for CGSE (which TS lacks) and the utility of always-valid inference.

To begin, we revisit live experiment 1 discussed in Section 4.2.1 and illustrated in Figure 4. Here, we show the daily empirical means for the arms in the TS and CGSE data in Figures 11a and 11b, respectively. As mentioned in Section 4.2.1, TS began to shift traffic to Arm 1 at day 26 before finally giving nearly all the traffic to this arm for the remainder of the experiment after day 30 despite it being clearly the worst performing arm. The reason for this behavior becomes more clear looking at Figure 11. Between days 22-25, TS gave almost no traffic to Arm 1. During this time period, the daily mean performance of each arm moves downward and consequently the posterior means move downward as well in TS for the arms receiving samples. However, the posterior mean for Arm 1 is unchanged since it did not receive samples. This results in Arm 1 starting to get more traffic between days 26-30. During this time period, the underlying performance of all arms move up significantly. This has the most impact on the posterior mean of Arm 1 in TS since it has begun to receive the most traffic. As this continues, the observations only reinforce the behavior of TS and this results in Arm 1 getting nearly all the traffic for the remainder of the experiment. This is a clear example of Simpson's paradox negatively impacting TS, while CGSE overcomes this issue by using the cumulative gain estimator and a traffic allocation that controls the variance of the estimator.

We observe a similar phenomenon in the experiment presented in Figure 12. In this experiment, CGSE produced a uniform traffic split over the arms during the experiment due to the small gaps and traffic. Yet, we observe that the arm that receives the most traffic in the TS

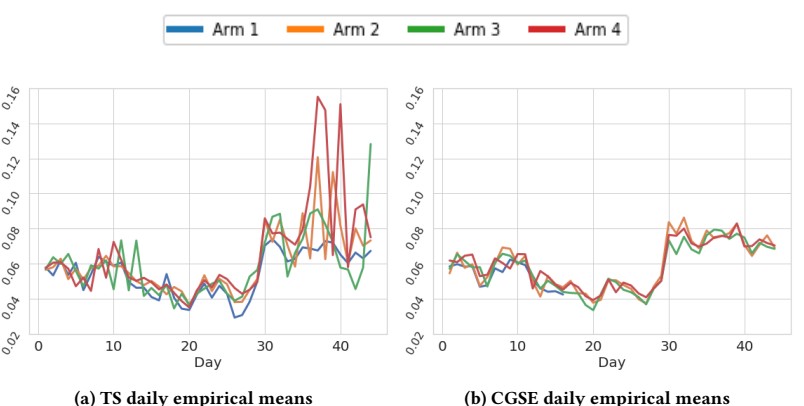

(a) TS daily empirical means

(b) CGSE daily empirical means

**Figure 11: Live Experiment 1 daily empirical means.**

data is in fact the arm with the lowest observed running empirical mean in the data from CGSE, indicating that the algorithm has made a mistake in terms of traffic allocation and this can also result in misleading inferences.

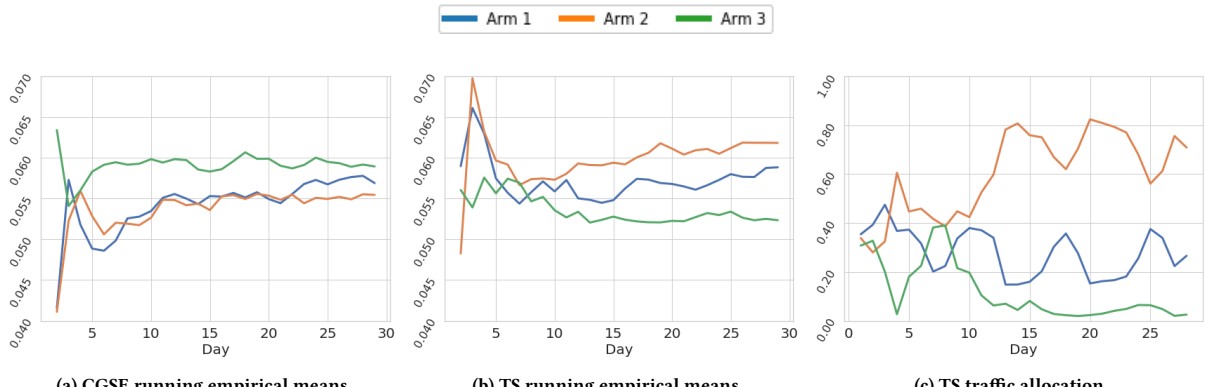

(a) CGSE running empirical means

(b) TS running empirical means

(c) TS traffic allocation

**Figure 12: Live Experiment 6: TS shifts the majority of traffic to the worst performing arm.**

We conclude with the experiment shown in Figure 13 where we see the weekly rolling empirical means over the arms and the running empirical means in a situation where no arm was eliminated until the final day of the experiment so the allocation was uniform. There is significant daily time-variation in the arm performances, yet the orange arm is almost always empirically best on a weekly rolling basis. This does not effect CGSE since it is similar to a constant gap setting and at the end of this experiment the optimal arm was identified as can be seen through the confidence intervals in Figure 13b. In contrast, for the TS, no decision could be made based on the data that was collected.

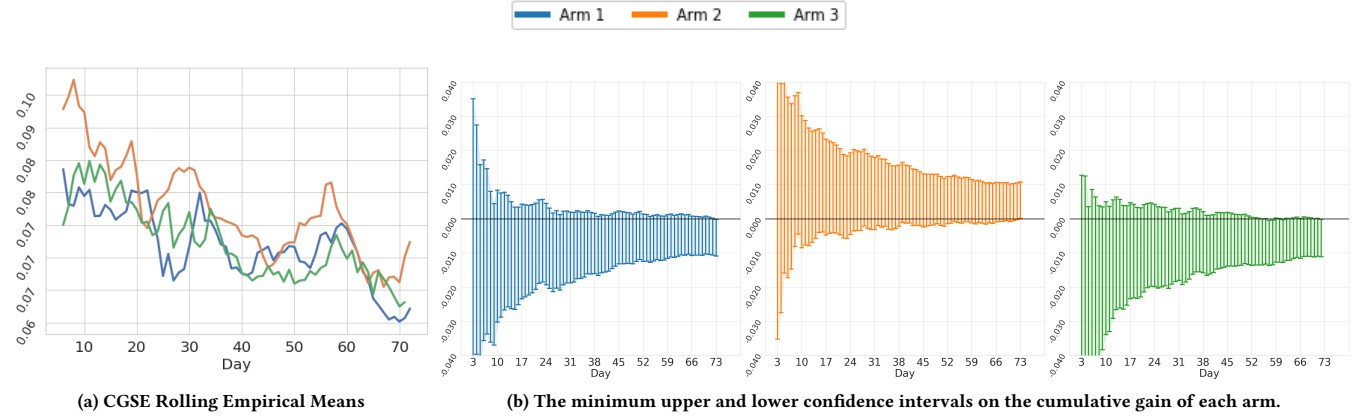

(a) CGSE Rolling Empirical Means

(b) The minimum upper and lower confidence intervals on the cumulative gain of each arm.

Figure 13: Live Experiment 7: CGSE identifies the optimal arm with significant non-stationarity.

