# OpenReview forum: "Best of Three Worlds: Adaptive Experimentation for Digital Marketing in Practice"
_ACM.org/TheWebConf/2024/Conference — TheWebConf24_

### Official Review · Reviewer_Dmx2 · 2023-11-02

**Novelty:** 4
**Technical Quality:** 5

**Review:**

**Summary**

This paper critically examines the application of Adaptive Experimental Design (AED) in industrial contexts, particularly its challenges under non-stationary conditions often present in real-world settings, contrasting this with traditional A/B/N testing methods. The authors highlight the limitations and potential misapplications of AED, especially in environments with dynamic feedback, through detailed case studies. They propose a novel AED framework that integrates a cumulative gain estimator with an always-valid inference mechanism and an elimination-based algorithmic approach. This system is designed to efficiently identify optimal treatment, minimize opportunity costs, and withstand common forms of time variation, offering a more reliable and cost-effective solution for practical experimentation scenarios. Theoretical guarantees and empirical evidence from production systems underscore the proposed system's robustness against the shortcomings of prevalent regret-minimizing algorithms.

**Strengths**
- The paper tackles the practically important problem relevant to online experimentation in industrial platform and environment non-stationarity

- The paper is well written and the arguments are easy to understand. In particular, the case studies in Section 2 are useful to grasp the key motivation of the work


- The paper succeeds in developing a fairly simple method that has guarantees and works empirically well in a rage of realistic non-stationary situations


- The paper's proposals are tested through extensive experiments on both offline and online setup. Experiment design and choice of baselines look mostly sufficient, and the provided results are promising and show the advantages of the proposed method, CGSE, under non-stationary environments

**Weaknesses**
- I was not sure how technically challenging and novel the provided guarantees are. They basically seem very simple statistical calculations or straightforward applications of known results


- The paper performs only real-world experiments, so the provided empirical results are specific to the authors’ applications. So, it would be better to also perform synthetic experiments where we can fully control the environment setup, making it possible to compare methods under a range of situations. For example, it would be interesting to compare methods when the number of arms is much larger and under varying levels of non-stationarity.


- This is very minor, but the title of the paper may be too broad. I was not sure what is done in the paper when looking at the current title

**Questions:**

No particular questions at the moment

**Reviewer Confidence:**

2: The reviewer is willing to defend the evaluation, but it is likely that the reviewer did not understand parts of the paper

**Scope:**

4: The work is relevant to the Web and to the track, and is of broad interest to the community

---

### Official Review · Reviewer_yb2F · 2023-11-22

**Novelty:** 4
**Technical Quality:** 4

**Review:**

This paper focuses on Adaptive experimental design (AED) which beyonds idealized stationary settings in traditional A/B testing methods. This paper introduces some important lessons learned regarding the challenges of naively using AED systems in industrial, and provides an effective approximation method.


Strengths:
1. This paper propose a practical method for real-world industry settings, and more importantly it provides theoretical guarantees about in time-varying situation, e.g., unbiasness, correct inference, stochastic environments & comparison.


2. The overall paper is easy to follow. Especially, the case study about the Simpson’s Paradox makes the readers realize the importance and motivation of the paper.

Weakness:

1. Your paper presents a range of insightful concepts and ideas, such as Valid Inference, Identify the Counterfactual, and the three worlds. However, I think your paper may be less accessible and unclear to broader WWW readers. I suggest the authors to link the methods to the concepts. I suggest you explicitly draw these connections, illustrating how your methods address or incorporate these concepts. This could be done through diagrams, flowcharts, or explicit explanations within the text.

2. The paper needs a separate “Related Works” section, to introduce previous AED methods and directly illustrate the differences between the proposed method and various related works.

3. I have some questions about the paper:
   - Since we need to consider a dynamic or time-varying situation, why not use reinforcement learning methods or collect a dataset to train ML model. I believe the patterns in this non-stationary case can be learned by using these methods.
   - How does Proposition 1 help in the paper? It is very similar to the selection bias method (inverse propensity) in recommender system which is very ideal and need to be approximated in practice.

**Questions:**

See Weakness part.

**Reviewer Confidence:**

3: The reviewer is confident but not certain that the evaluation is correct

**Scope:**

3: The work is somewhat relevant to the Web and to the track, and is of narrow interest to a sub-community

---

### Official Review · Reviewer_HuxD · 2023-11-22

**Novelty:** 5
**Technical Quality:** 5

**Review:**

The paper aims at improving AED by cumulative gain estimator with always valid inference and an elimination-based algorithmic approach. The paper is well written, easy to follow, Idea of the paper, discussing actual research topic. I like the idea of the proposal, despite the fact that I have several comments.

Section 2 is written in a story-like manner, which can be greatly reduced. It is somehow combined with the related work resulting in some hybrid section. I lack the clear and up-to-date state-of-the-art comparison (two references from 2022, one from 2021, rest is older).
Several claims in the paper are not sufficiently substantiated, sometimes the argumentation follows “trust us, we have experience“ (e.g., end of section 3).

Last but not least, the evaluation section lacks details on the dataset, which greatly degrades the reproducibility of the paper. Why any standard dataset wasn’t used?

Overall, I suggest supporting claims across the paper with the evidence and to increase paper reproducibility.

**Questions:**

no

**Ethics Review Description:**

nan

**Reviewer Confidence:**

2: The reviewer is willing to defend the evaluation, but it is likely that the reviewer did not understand parts of the paper

**Scope:**

3: The work is somewhat relevant to the Web and to the track, and is of narrow interest to a sub-community

---

### Official Review · Reviewer_WZBL · 2023-11-24

**Novelty:** 5
**Technical Quality:** 5

**Review:**

Summary:
This work proposes a framework for Adaptive Experimental Design (AED) for counterfactual inference based on real-world industrial experiences of using AED systems. In particular, the proposed framework identifies the counterfactual optimal treatment efficiently, mitigates opportunity cost and is robust to time variation usually observed in industrial settings. The paper provides empirical evidence as well as theoretical guarantees in support of their proposed framework.

Paper Strength:
The paper highlights important/significant problems related to real-world deployments of AEDs (mainly, related to inference and non-stationarity). The proposed framework is rigorous and robust, backed by empirical and theoretical guarantees.
The paper is well-written and easy to follow.

However, I have some concerns with lack of relation to existing literature and lack of a discussion section in the paper.

Paper Weakness:
The paper does a great job motivating the problems of inference and non-stationarity in bandit settings with case studies. However, the paper would benefit from highlighting gaps in existing literature (especially within Web Conference) that have attempted to solve these problems. Some papers that might be useful to look at:
Deliu, Nina, Joseph J. Williams, and Sofia S. Villar. "Efficient inference without trading-off regret in bandits: An allocation probability test for Thompson sampling." arXiv preprint arXiv:2111.00137 (2021).
Trovo, Francesco, et al. "Sliding-window thompson sampling for non-stationary settings." Journal of Artificial Intelligence Research 68 (2020): 311-364.
Wu, Qingyun, et al. "Dynamic ensemble of contextual bandits to satisfy users' changing interests." The World Wide Web Conference. 2019.

The paper is missing a discussion tying the paper’s findings with existing literature, future work, and limitations of the existing work.

**Questions:**

How is the proposed framework situated within the broader line of work related to adaptive experimental design within the Web Conference literature? How is it situated within the literature of adaptive experimental designs?

**Reviewer Confidence:**

2: The reviewer is willing to defend the evaluation, but it is likely that the reviewer did not understand parts of the paper

**Scope:**

4: The work is relevant to the Web and to the track, and is of broad interest to the community

---

### Official Review · Reviewer_UT6o · 2023-11-28

**Novelty:** 5
**Technical Quality:** 4

**Review:**

In this work, the authors discuss different experimental approaches, algorithms employed and how they might affect the conclusions made upon experiments completion. Specifically, it deals with the type of A/B experiments in which the traffic is dynamically allocated during the experiment. the authors propose to use cumulative gain for drawing reliable conclusions about the results. The method has been evaluated offline and online with four experiments.

The paper is well written, although quite theoretical hence challenging to grasp overall. The topic is extremely relevant, as it deals with how we make conclusions based on (online) experiments, hence the decisions about different web dimensions that we employ. My main concerns about the work is that the proposed approach is evaluated (with the online experiments) only against the most conservative approach, that is TS (Thompson Sampling), and not the other mentioned approaches. To this end, my questions for the authors are as follows:
1. Why not comparing the proposed approach against other approaches, i.e., TTTS, Uniform and BOB?
2. Can it actually happen that Simpson's paradox, when it happens, actually be a sign that we should have a deeper look in whether some contextual information is influencing our experiment results, that at the beginning we did not properly consider? Hence, by disregarding this information, by overlooking with other metrics (as cumulative gain) are we actually loosing?

**Questions:**

1. Why not comparing the proposed approach against other approaches, i.e., TTTS, Uniform and BOB?
2. Can it actually happen that Simpson's paradox, when it happens, actually be a sign that we should have a deeper look in whether some contextual information is influencing our experiment results, that at the beginning we did not properly consider? Hence, by disregarding this information, by overlooking with other metrics (as cumulative gain) are we actually loosing?

**Ethics Review Description:**

-

**Reviewer Confidence:**

2: The reviewer is willing to defend the evaluation, but it is likely that the reviewer did not understand parts of the paper

**Scope:**

4: The work is relevant to the Web and to the track, and is of broad interest to the community

---

### Decision · Program_Chairs · 2024-01-22

**Decision:**

Accept

**Comment:**

This paper presents empirical observations on adaptive experiment design in an industrial setting, where the traffic for A/B test was constructed adaptively. The authors also presented a solution for locating counterfactual optimal treatments, based on a cumulative gain estimator, which is robust to time variations in practical situations.

 All reviewers agreed that the paper is very well-written and easy to follow, the proposed solution is neat and practically effective for an important real-world problem. The reviewers also suggested some baselines and references for the authors to compare and tailor the technical content a bit to WWW audiences. But in general, there is no strong opposition against this work.

 As a result, we are happy to recommend accepting this work, but we still suggest the authors to further polish the paper's content according to the reviewers' comments and suggestions to make the content more accessible to the WWW community.